# THE IMPORTANCE OF PESSIMISM IN FIXED-DATASET POLICY OPTIMIZATION

## ABSTRACT

We study worst-case guarantees on the expected return of fixed-dataset policy optimization algorithms. Our core contribution is a unified conceptual and mathematical framework for the study of algorithms in this regime. This analysis reveals that for naïve approaches, the possibility of erroneous value overestimation leads to a difficult-to-satisfy requirement: in order to guarantee that we select a policy which is near-optimal, we may need the dataset to be informative of the value of every policy. To avoid this, algorithms can follow the *pessimism principle*, which states that we should choose the policy which acts optimally in the worst possible world. We show why pessimistic algorithms can achieve good performance even when the dataset is not informative of every policy, and derive families of algorithms which follow this principle. These theoretical findings are validated by experiments on a tabular gridworld, and deep learning experiments on four MinAtar environments.

## 1 INTRODUCTION

We consider *fixed-dataset policy optimization* (FDPO), in which a dataset of transitions from an environment is used to find a policy with high return.[1] We compare FDPO algorithms by their worst-case performance, expressed as high-probability guarantees on the suboptimality of the learned policy. It is perhaps obvious that in order to maximize worst-case performance, a good FDPO algorithm should select a policy with high worst-case value. We call this the *pessimism principle* of exploitation, as it is analogous to the widely-known *optimism principle* (Lattimore & Szepesvári, 2020) of exploration.[2]

Our main contribution is a theoretical justification of the pessimism principle in FDPO, based on a bound that characterizes the suboptimality incurred by an FDPO algorithm. We further demonstrate how this bound may be used to derive principled algorithms. Note that the core novelty of our work is not the idea of pessimism, which is an intuitive concept that appears in a variety of contexts; rather, our contribution is a set of theoretical results rigorously explaining *how* pessimism is important in the specific setting of FDPO. An example conveying the intuition behind our results can be found in Appendix G.1.

We first analyze a family of non-pessimistic *naïve FDPO algorithms*, which estimate the environment from the dataset via maximum likelihood and then apply standard dynamic programming techniques. We prove a bound which shows that the worst-case suboptimality of these algorithms is guaranteed to be small when the dataset contains enough data that we are certain about the value of every possible policy. This is caused by the outsized impact of value overestimation errors on suboptimality, sometimes called the optimizer's curse (Smith & Winkler, 2006). It is a fundamental consequence of ignoring the disconnect between the true environment and the picture painted by our limited observations. Importantly, it is not reliant on errors introduced by function approximation.

---

[1] We use the term fixed-dataset policy optimization to emphasize the computational procedure; this setting has also been referred to as *batch RL* (Ernst et al., 2005; Lange et al., 2012) and more recently, *offline RL* (Levine et al., 2020). We emphasize that this is a well-studied setting, and we are simply choosing to refer to it by a more descriptive name.

[2] The optimism principle states that we should select a policy with high best-case value.

We contrast these findings with an analysis of *pessimistic FDPO algorithms*, which select a policy that maximizes some notion of worst-case expected return. We show that these algorithms do not require datasets which inform us about the value of every policy to achieve small suboptimality, due to the critical role that pessimism plays in preventing overestimation. Our analysis naturally leads to two families of principled pessimistic FDPO algorithms. We prove their improved suboptimality guarantees, and confirm our claims with experiments on a gridworld.

Finally, we extend one of our pessimistic algorithms to the deep learning setting. Recently, several deep-learning-based algorithms for fixed-dataset policy optimization have been proposed (Agarwal et al., 2019; Fujimoto et al., 2019; Kumar et al., 2019; Laroche et al., 2019; Jaques et al., 2019; Kidambi et al., 2020; Yu et al., 2020; Wu et al., 2019; Wang et al., 2020; Kumar et al., 2020; Liu et al., 2020). Our work is complementary to these results, as our contributions are conceptual, rather than algorithmic. Our primary goal is to theoretically unify existing approaches and motivate the design of pessimistic algorithms more broadly. Using experiments in the MinAtar game suite (Young & Tian, 2019), we provide empirical validation for the predictions of our analysis.

The problem of fixed-dataset policy optimization is closely related to the problem of reinforcement learning, and as such, there is a large body of work which contains ideas related to those discussed in this paper. We discuss these works in detail in Appendix E.

## 2 BACKGROUND

We anticipate most readers will be familiar with the concepts and notation, which is fairly standard in the reinforcement learning literature. In the interest of space, we relegate a full presentation to Appendix A. Here, we briefly give an informal overview of the background necessary to understand the main results.

We represent the environment as a *Markov Decision Process (MDP)*, denoted $\mathcal{M} := \langle \mathcal{S}, \mathcal{A}, \mathcal{R}, P, \gamma, \rho \rangle$. We assume without loss of generality that $\mathcal{R}(\langle s, a \rangle) \in [0, 1]$, and denote its expectation as $\mathbf{r}(\langle s, a \rangle)$. $\rho$ represents the start-state distribution. Policies $\pi$ can act in the environment, represented by action matrix $A^\pi$, which maps each state to the probability of each state-action when following $\pi$. Value functions $\mathbf{v}$ assign some real value to each state. We use $\mathbf{v}_{\mathcal{M}}^\pi$ to denote the value function which assigns the sum of discounted rewards in the environment when following policy $\pi$. A dataset $D$ contains transitions sampled from the environment. From a dataset, we can compute the empirical reward and transition functions, $\mathbf{r}_D$ and $P_D$, and the empirical policy, $\hat{\pi}_D$.

An important concept for our analysis is the *value uncertainty function*, denoted $\boldsymbol{\mu}_{D,\delta}^\pi$, which returns a high-probability upper-bound to the error of a value function derived from dataset $D$. Certain value uncertainty functions are *decomposable* by states or state-actions, meaning they can be written as the weighted sum of more local uncertainties. See Appendix B for more detail.

Our goal is to analyze the suboptimality of a specific class of FDPO algorithms, called *value-based FDPO algorithms*, which have a straightforward structure: they use a *fixed-dataset policy evaluation (FDPE)* algorithm to assign a value to each policy, and then select the policy with the maximum value. Furthermore, we consider FDPE algorithms whose solutions satisfy a fixed-point equation. Thus, a fixed-point equation defines a FDPE objective, which in turn defines a value-based FDPO objective; we call the set of all algorithms that implement these objectives the *family* of algorithms defined by the fixed-point equation.

## 3 OVER/UNDER DECOMPOSITION OF SUBOPTIMALITY

Our first theoretical contribution is a simple but informative bound on the suboptimality of any value-based FDPO algorithm. Next, in Section 4, we make this concrete by defining the family of naïve algorithms and invoking this bound. This bound is insightful because it distinguishes the impact of errors of value overestimation from errors of value underestimation, defined as:

**Definition 1.** *Consider any fixed-dataset policy evaluation algorithm $\mathcal{E}$ on any dataset $D$ and any policy $\pi$. Denote $\mathbf{v}_D^\pi := \mathcal{E}(D, \pi)$. We define the* underestimation error *as* $\mathbb{E}_\rho[\mathbf{v}_{\mathcal{M}}^\pi - \mathbf{v}_D^\pi]$ *and* overestimation error *as* $\mathbb{E}_\rho[\mathbf{v}_D^\pi - \mathbf{v}_{\mathcal{M}}^\pi]$.

The following lemma shows how these quantities can be used to bound suboptimality.

**Lemma 1** (Value-based FDPO suboptimality bound). *Consider any value-based fixed-dataset policy optimization algorithm $\mathcal{O}^{VB}$, with fixed-dataset policy evaluation subroutine $\mathcal{E}$. For any policy $\pi$ and dataset $D$, denote $\mathbf{v}_D^\pi := \mathcal{E}(D, \pi)$. The suboptimality of $\mathcal{O}^{VB}$ is bounded by*

$$\text{SUBOPT}(\mathcal{O}^{VB}(D)) \leq \inf_\pi \left( \mathbb{E}_\rho[\mathbf{v}_\mathcal{M}^{\pi_\mathcal{M}^*} - \mathbf{v}_\mathcal{M}^\pi] + \mathbb{E}_\rho[\mathbf{v}_\mathcal{M}^\pi - \mathbf{v}_D^\pi] \right) + \sup_\pi \left( \mathbb{E}_\rho[\mathbf{v}_D^\pi - \mathbf{v}_\mathcal{M}^\pi] \right)$$

*Proof.* See Appendix C.1. $\qquad\square$

This bound is tight; see Appendix C.2. The bound highlights the potentially outsized impact of overestimation on the suboptimality of a FDPO algorithm. To see this, we consider each of its terms in isolation:

$$\text{SUBOPT}(\mathcal{O}^{VB}(D)) \leq \inf_\pi \Big( \underbrace{\overbrace{\mathbb{E}_\rho[\mathbf{v}_\mathcal{M}^{\pi_\mathcal{M}^*} - \mathbf{v}_\mathcal{M}^\pi]}^{(\text{A}1)} + \overbrace{\mathbb{E}_\rho[\mathbf{v}_\mathcal{M}^\pi - \mathbf{v}_D^\pi]}^{(\text{A}2)}}_{(\text{A})} \Big) + \sup_\pi \Big( \underbrace{\overbrace{\mathbb{E}_\rho[\mathbf{v}_D^\pi - \mathbf{v}_\mathcal{M}^\pi]}^{(\text{B}1)}}_{(\text{B})} \Big)$$

The term labeled (A) reflects the degree to which the dataset informs us of a near-optimal policy. For any policy $\pi$, (A1) captures the suboptimality of that policy, and (A2) captures its underestimation error. Since (A) takes an infimum, this term will be small whenever there is at least one reasonable policy whose value is not very underestimated.

On the other hand, the term labeled (B) corresponds to the largest overestimation error on any policy. Because it consists of a supremum over all policies, it will be small only when *no policies are overestimated at all*. Even a single overestimation can lead to significant suboptimality.

We see from these two terms that errors of overestimation and underestimation have differing impacts on suboptimality, suggesting that algorithms should be designed with this asymmetry in mind. We will see in Section 5 how this may be done. But first, let us further understand why this is necessary by studying in more depth a family of algorithms which treats its errors of overestimation and underestimation equivalently.

## 4 NAÏVE ALGORITHMS

The goal of this section is to paint a high-level picture of the worst-case suboptimality guarantees of a specific family of non-pessimistic approaches, which we call *naïve FDPO algorithms*. Informally, the naïve approach is to take the limited dataset of observations at face value, treating it as though it paints a fully accurate picture of the environment. Naïve algorithms construct a maximum-likelihood MDP from the dataset, then use standard dynamic programming approaches on this empirical MDP.

**Definition 2.** *A* naïve algorithm *is any algorithm in the family defined by the fixed-point function*

$$f_{naïve}(\mathbf{v}^\pi) := A^\pi(\mathbf{r}_D + \gamma P_D \mathbf{v}^\pi).$$

Various FDPE and FDPO algorithms from this family could be described; in this work, we do not study these implementations in detail, although we do give pseudocode for some implementations in Appendix D.1.

One example of a naïve FDPO algorithm which can be found in the literature is certainty equivalence (Jiang, 2019a). The core ideas behind naïve algorithms can also be found in the function approximation literature, for example in FQI (Ernst et al., 2005; Jiang, 2019b). Additionally, when available data is held fixed, nearly all existing deep reinforcement learning algorithms are transformed into naïve value-based FDPO algorithms. For example, DQN (Mnih et al., 2015) with a fixed replay buffer is a naïve value-based FDPO algorithm.

**Theorem 1** (Naïve FDPO suboptimality bound). *Consider any naïve value-based fixed-dataset policy optimization algorithm $\mathcal{O}_{naïve}^{VB}$. Let $\boldsymbol{\mu}$ be any value uncertainty function. With probability at least $1 - \delta$, the suboptimality of $\mathcal{O}_{naïve}^{VB}$ is bounded with probability at least $1 - \delta$ by*

$$\text{SUBOPT}(\mathcal{O}_{naïve}^{VB}(D)) \leq \inf_\pi \left( \mathbb{E}_\rho[\mathbf{v}_\mathcal{M}^{\pi_\mathcal{M}^*} - \mathbf{v}_\mathcal{M}^\pi] + \mathbb{E}_\rho[\boldsymbol{\mu}_{D,\delta}^\pi] \right) + \sup_\pi \mathbb{E}_\rho[\boldsymbol{\mu}_{D,\delta}^\pi]$$

*Proof.* This result follows directly from Lemma 1 and Lemma 3. □

The infimum term is small whenever there is some reasonably good policy with low value uncertainty. In practice, this condition can typically be satisfied, for example by including expert demonstrations in the dataset. On the other hand, the supremum term will only be small if we have low value uncertainty for all policies – a much more challenging requirement. This explains the behavior of pathological examples, e.g. in Appendix G.1, where performance is poor despite access to virtually unlimited amounts of data from a near-optimal policy. Such a dataset ensures that the first term will be small by reducing value uncertainty of the near-optimal data collection policy, but does little to reduce the value uncertainty of any other policy, leading the second term to be large.

However, although pathological examples exist, it is clear that this bound will not be tight on all environments. It is reasonable to ask: is it likely that this bound will be tight on real-world examples? We argue that it likely will be. We identify two properties that most real-world tasks share: (1) The set of policies is pyramidal: there are an enormous number of bad policies, many mediocre policies, a few good policies, etc. (2) Due to the size of the state space and cost of data collection, most policies have high value uncertainty.

Given that these assumptions hold, naïve algorithms will perform as poorly on most real-world environments as they do on pathological examples. Consider: there are many more policies than there is data, so there will be many policies with high value uncertainty; naïve algorithms will likely overestimate several of these policies, and erroneously select one; since good policies are rare, the selected policy will likely be bad. It follows that running naïve algorithms on real-world problems will typically yield suboptimality close to our worst-case bound. And, indeed, on deep RL benchmarks, which are selected due to their similarity to real-world settings, overestimation has been widely observed, typically correlated with poor performance (Bellemare et al., 2016; Van Hasselt et al., 2016; Fujimoto et al., 2019).

## 5 The Pessimism Principle

*"Behave as though the world was plausibly worse than you observed it to be."* The pessimism principle tells us how to exploit our current knowledge to find the stationary policy with the best worst-case guarantee on expected return. We consider two specific families of pessimistic algorithms, the *uncertainty-aware pessimistic algorithms* and *proximal pessimistic algorithms*, and bound the worst-case suboptimality of each. These algorithms each include a hyperparameter, $\alpha$, controlling the amount of pessimism, interpolating from fully-naïve to fully-pessimistic. (For a discussion of the implications of the latter extreme, see Appendix G.2.) Then, we will compare the two families, and see how the proximal family is simply a trivial special case of the more general uncertainty-aware family of methods.

### 5.1 Uncertainty-Aware Pessimistic Algorithms

Our first family of pessimistic algorithms is the *uncertainty-aware (UA) pessimistic algorithms*. As the name suggests, this family of algorithms estimates the state-wise Bellman uncertainty and penalizes policies accordingly, leading to a pessimistic value estimate and a preference for policies with low value uncertainty.

**Definition 3.** *An* uncertainty-aware pessimistic algorithm*, with a Bellman uncertainty function* $\mathbf{u}_{D,\delta}^{\pi}$ *and pessimism hyperparameter* $\alpha \in [0,1]$*, is any algorithm in the family defined by the fixed-point function*

$$f_{ua}(\mathbf{v}^{\pi}) = A^{\pi}(\mathbf{r}_D + \gamma P_D \mathbf{v}^{\pi}) - \alpha \mathbf{u}_{D,\delta}^{\pi}$$

This fixed-point function is simply the naïve fixed-point function penalized by the Bellman uncertainty. This can be interpreted as being pessimistic about the outcome of every action. Note that it remains to specify a technique to compute the Bellman uncertainty function, e.g. Appendix B.1, in order to get a concrete algorithm. It is straightforward to construct algorithms from this family by modifying naïve algorithms to subtract the penalty term. Similar algorithms have been explored in the safe RL literature (Ghavamzadeh et al., 2016; Laroche et al., 2019) and the robust MDP literature (Givan et al., 1997), where algorithms with high-probability performance guarantees are useful in the context of ensuring safety.

**Theorem 2** (Uncertainty-aware pessimistic FDPO suboptimality bound). *Consider an uncertainty-aware pessimistic value-based fixed-dataset policy optimization algorithm $\underline{\mathcal{Q}}_{ua}^{VB}$. Let $\mathbf{u}_{D,\delta}^{\pi}$ be any Bellman uncertainty function, $\boldsymbol{\mu}_{D,\delta}^{\pi}$ be a corresponding value uncertainty function, and $\alpha \in [0,1]$ be any pessimism hyperparameter. The suboptimality of $\underline{\mathcal{Q}}_{ua}^{VB}$ is bounded with probability at least $1 - \delta$ by*

$$\textsc{SubOpt}(\underline{\mathcal{Q}}_{ua}^{VB}(D)) \leq \inf_{\pi} \left( \mathbb{E}_{\rho}[\mathbf{v}_{\mathcal{M}}^{\pi_{\mathcal{M}}^{*}} - \mathbf{v}_{\mathcal{M}}^{\pi}] + (1+\alpha) \cdot \mathbb{E}_{\rho}[\boldsymbol{\mu}_{D,\delta}^{\pi}] \right) + (1-\alpha) \cdot \left( \sup_{\pi} \mathbb{E}_{\rho}[\boldsymbol{\mu}_{D,\delta}^{\pi}] \right)$$

*Proof.* See Appendix C.7. □

This bound should be contrasted with our result from Theorem 1. With $\alpha = 0$, the family of pessimistic algorithms reduces to the family of naïve algorithms, so the bound is correspondingly identical. We can add pessimism by increasing $\alpha$, and this corresponds to a decrease in the magnitude of the supremum term. When $\alpha = 1$, there is no supremum term at all. In general, the optimal value of $\alpha$ lies between the two extremes.

To further understand the power of this approach, it is illustrative to compare it to imitation learning. Consider the case where the dataset contains a small number of expert trajectories but also a large number of interactions from a random policy, i.e. when learning from suboptimal demonstrations (Brown et al., 2019). If the dataset contained *only* a small amount of expert data, then both an UA pessimistic FDPO algorithm and an imitation learning algorithm would return a high-value policy. However, the injection of sufficiently many random interactions would degrade the performance of imitation learning algorithms, whereas UA pessimistic algorithms would continue to behave similarly to the expert data.

## 5.2 PROXIMAL PESSIMISTIC ALGORITHMS

The next family of algorithms we study are the *proximal pessimistic algorithms*, which implement pessimism by penalizing policies that deviate from the empirical policy. The name *proximal* was chosen to reflect the idea that these algorithms prefer policies which stay "nearby" to the empirical policy. Many FDPO algorithms in the literature, and in particular several recently-proposed deep learning algorithms (Fujimoto et al., 2019; Kumar et al., 2019; Laroche et al., 2019; Jaques et al., 2019; Wu et al., 2019; Liu et al., 2020), resemble members of the family of proximal pessimistic algorithms; see Appendix E. Also, another variant of the proximal pessimistic family, which uses state density instead of state-conditional action density, can be found in Appendix C.9.

**Definition 4.** *A proximal pessimistic algorithm with pessimism hyperparameter $\alpha \in [0,1]$ is any algorithm in the family defined by the fixed-point function*

$$f_{proximal}(\mathbf{v}^{\pi}) = A^{\pi}(\mathbf{r}_D + \gamma P_D \mathbf{v}^{\pi}) - \alpha \left( \frac{TV_{\mathcal{S}}(\pi, \hat{\pi}_D)}{(1-\gamma)^2} \right)$$

**Theorem 3** (Proximal pessimistic FDPO suboptimality bound). *Consider any proximal pessimistic value-based fixed-dataset policy optimization algorithm $\underline{\mathcal{Q}}_{proximal}^{VB}$. Let $\boldsymbol{\mu}$ be any state-action-wise decomposable value uncertainty function, and $\alpha \in [0,1]$ be a pessimism hyperparameter. For any dataset $D$, the suboptimality of $\underline{\mathcal{Q}}_{proximal}^{VB}$ is bounded with probability at least $1 - \delta$ by*

$$\textsc{SubOpt}(\mathcal{O}_{proximal}(D)) \leq \inf_{\pi} \left( \mathbb{E}_{\rho}[\mathbf{v}_{\mathcal{M}}^{\pi_{\mathcal{M}}^{*}} - \mathbf{v}_{\mathcal{M}}^{\pi}] + \mathbb{E}_{\rho} \left[ \boldsymbol{\mu}_{D,\delta}^{\pi} + \alpha(I - \gamma A^{\pi} P_D)^{-1} \left( \frac{TV_{\mathcal{S}}(\pi, \hat{\pi}_D)}{(1-\gamma)^2} \right) \right] \right)$$

$$+ \sup_{\pi} \left( \mathbb{E}_{\rho} \left[ \boldsymbol{\mu}_{D,\delta}^{\pi} - \alpha(I - \gamma A^{\pi} P_D)^{-1} \left( \frac{TV_{\mathcal{S}}(\pi, \hat{\pi}_D)}{(1-\gamma)^2} \right) \right] \right)$$

*Proof.* See Appendix C.8. □

Once again, we see that as $\alpha$ grows, the large supremum term shrinks; similarly, by Lemma 5, when we have $\alpha = 1$, the supremum term is guaranteed to be non-positive.[3] The primary limitation of

---

[3] Initially, it will contain $\boldsymbol{\mu}_{D,\delta}^{\pi'}$, but this can be removed since it is not dependent on $\pi$.

the proximal approach is the looseness of the value lower-bound. Intuitively, this algorithm can be understood as performing imitation learning, but permitting minor deviations. Constraining the policy to be near in distribution to the empirical policy can fail to take advantage of highly-visited states which are reached via many trajectories. In fact, in contrast to both the naïve approach and the UA pessimistic approach, in the limit of infinite data this approach is not guaranteed to converge to the optimal policy. Also, note that when $\alpha \geq 1 - \gamma$, this algorithm is identical to imitation learning.

## 5.3 THE RELATIONSHIP BETWEEN UNCERTAINTY-AWARE AND PROXIMAL ALGORITHMS

Though these two families may appear on the surface to be quite different, they are in fact closely related. A key insight of our theoretical work is that it reveals the important connection between these two approaches. Concretely: proximal algorithms are uncertainty-aware algorithms which use a trivial value uncertainty function.

To see this, we show how to convert an uncertainty-aware penalty into a proximal penalty. let $\boldsymbol{\mu}$ be any state-action-wise decomposable value uncertainty function. For any dataset $D$, we have

$$\boldsymbol{\mu}_{D,\delta}^{\pi} = \boldsymbol{\mu}_{D,\delta}^{\hat{\pi}_D} + (I - \gamma A^{\pi} P_D)^{-1} \left( (A^{\pi} - A^{\hat{\pi}_D})(\mathbf{u}_{D,\delta} + \gamma P_D \boldsymbol{\mu}_{D,\delta}^{\hat{\pi}_D}) \right) \qquad \text{(Lemma 4)}$$

$$\leq \boldsymbol{\mu}_{D,\delta}^{\hat{\pi}_D} + (I - \gamma A^{\pi} P_D)^{-1} \left( \mathbf{TV}_{\mathcal{S}}(\pi, \pi') \left( \frac{1}{(1-\gamma)^2} \right) \right). \qquad \text{(Lemma 5)}$$

We began with the uncertainty penalty. In the first step, we rewrote the uncertainty for $\pi$ into the sum of two terms: the uncertainty for $\hat{\pi}_D$, and the difference in uncertainty between $\pi$ and $\hat{\pi}_D$ on various actions. In the second step, we chose our state-action-wise Bellman uncertainty to be $\frac{1}{1-\gamma}$, which is a trivial upper bound; we also upper-bound the signed policy difference with the total variation. This results in the proximal penalty.[4]

Thus, we see that proximal penalties are equivalent to uncertainty-aware penalties which use a specific, trivial uncertainty function. This result suggests that uncertainty-aware algorithms are strictly better than their proximal counterparts. There is no looseness in this result: for any proximal penalty, we will always be able to find a tighter uncertainty-aware penalty by replacing the trivial uncertainty function with something tighter.

However, currently, proximal algorithms are quite useful in the context of deep learning. This is because the only uncertainty function that can currently be implemented for neural networks is the trivial uncertainty function. Until we discover how to compute uncertainties for neural networks, proximal pessimistic algorithms will remain the only theoretically-motivated family of algorithms.

## 6 EXPERIMENTS

We implement algorithms from each family to empirically investigate whether their performance of follows the predictions of our bounds. Below, we summarize the key predictions of our theory.

- **Imitation.** This algorithm simply learns to copy the empirical policy. It performs well if and only if the data collection policy performs well.
- **Naïve.** This algorithm performs well only when almost no policies have high value uncertainty. This means that when the data is collected from any mostly-deterministic policy, performance of this algorithm will be poor, since many states will be missing data. Stochastic data collection improves performance. As the size of the dataset grows, this algorithm approaches optimality.
- **Uncertainty-aware.** This algorithm performs well when there is data on states visited by near-optimal policies. This is the case when a small amount of data has been collected from a near-optimal policy, or a large amount of data has been collected from a worse policy. As the size of the dataset grows, this algorithm to approaches optimality. This approach outperforms all other approaches.

---

[4]When constructing the penalty, we can ignore the first term, which does not contain $\pi$, and so is irrelevant to optimization.

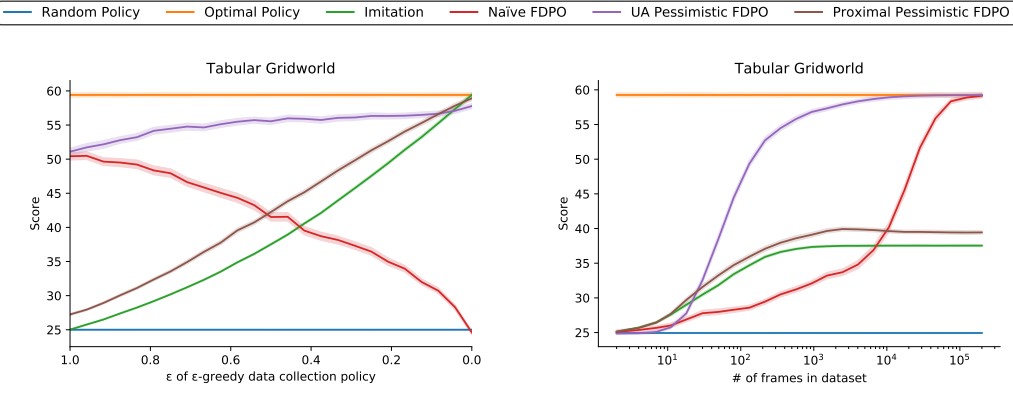

(a) Performance of FDPO algorithms on a dataset of 2000 transitions, as the data collection policy is interpolated from random to optimal.

(b) Performance of FDPO algorithms as dataset size increases. Data is collected with an optimal $\epsilon$-greedy policy, with $\epsilon = 50\%$.

Figure 1: Tabular gridworld experiments.

- **Proximal.** This algorithm roughly mirrors the performance of the imitation approach, but improves upon it. As the size of the dataset grows, this algorithm does not approach optimality, as the penalty persists even when the environment's dynamics are perfectly captured by the dataset.

Our experimental results qualitatively align with our predictions in both the tabular and deep learning settings, giving evidence that the picture painted by our theoretical analysis truly describes the FDPO setting. See Appendix D for pseudocode of all algorithms; see Appendix F for details on the experimental setup; see Appendix G.3 for additional experimental considerations for deep learning experiments that will be of interest to practicioners. For an open-source implementation, including full details suitable for replication, please refer to the code in the accompanying GitHub repository: `github.com/anonymized`

**Tabular.** The first tabular experiment, whose results are shown in Figure 1(a), compares the performance of the algorithms as the policy used to collect the dataset is interpolated from the uniform random policy to an optimal policy using $\epsilon$-greedy. The second experiment, whose results are shown in Figure 1(b), compares the performance of the algorithms as we increase the size of the dataset from 1 sample to 200000 samples. In both experiments, we notice a qualitative difference between the trends of the various algorithms, which aligns with the predictions of our theory.

**Neural network.** The results of these experiments can be seen in Figure 2. Similarly to the tabular experiments, we see that the naïve approach performs well when data is fully exploratory, and poorly when data is collected via an optimal policy; the pure imitation approach performs better when the data collection policy is closer to optimal. The pessimistic approach achieves the best of both worlds: it correctly imitates a near-optimal policy, but also learns to improve upon it somewhat when the data is more exploratory. One notable failure case is in FREEWAY, where the performance of the pessimistic approach barely improves upon the imitation policy, despite the naïve approach performing near-optimally for intermediate values of $\epsilon$.

## 7  DISCUSSION AND CONCLUSION

In this work, we provided a conceptual and mathematical framework for thinking about fixed-dataset policy optimization. Starting from intuitive building blocks of uncertainty and the over-under decomposition, we showed the core issue with naïve approaches, and introduced the pessimism principle as the defining characteristic of solutions. We described two families of pessimistic algorithms, uncertainty-aware and proximal. We see theoretically that both of these approaches have advantages over the naïve approach, and observed these advantages empirically. Comparing these two families of pessimistic algorithms, we see both theoretically and empirically that uncertainty-aware

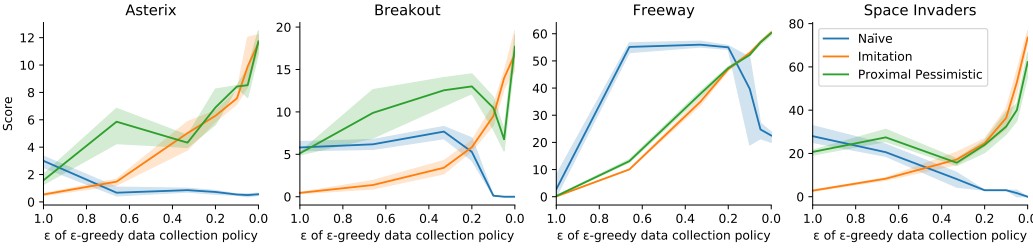

Figure 2: Performance of deep FDPO algorithms on a dataset of 500000 transitions, as the data collection policy is interpolated from near-optimal to random. Note that here, the only pessimistic algorithm evaluated is proximal.

algorithms are strictly better than proximal algorithms, and that proximal algorithms may not yield the optimal policy, even with infinite data.

**Future directions.** Our results indicate that research in FDPO should not focus on proximal algorithms. The development of neural uncertainty estimation techniques will enable principled uncertainty-aware deep learning algorithms. As is evidenced by our tabular results, we expect these approaches to yield dramatic performance improvements, rendering algorithms derived from the proximal family (Kumar et al., 2019; Fujimoto et al., 2019; Laroche et al., 2019; Kumar et al., 2020) obsolete.

**On ad-hoc solutions.** It is undoubtably disappointing to see that proximal algorithms, which are far easier to implement, are fundamentally limited in this way. It is tempting to propose various ad-hoc solutions to mitigate the flaws of proximal pessimistic algorithms in practice. However, in order to ensure that the resulting algorithm is principled, one must be careful. For example, one might consider tuning $\alpha$; however, doing the tuning requires evaluating each policy in the environment, which involves gaining information by interacting with the environment, which is not permitted by the problem setting. Or, one might consider e.g. an adaptive pessimism hyperparameter which decays with the size of the dataset; however, in order for such a penalty to be principled, it must be based on an uncertainty function, at which point we may as well just use an uncertainty-aware algorithm.

**Stochastic policies.** One surprising property of pessimsitic algorithms is that the optimal policy is often stochastic. This is because the penalty term included in their fixed-point objective is often minimized by stochastic policies. For the penalty of proximal pessimistic algorithms, it is easy to see that this will be the case for any non-deterministic empirical policy; for UA pessimsitic algorithms, it is dependent on the choice of Bellman uncertainty function, but often still holds (see Appendix B.2 for the derivation of a Bellman uncertainty function with this property). This observation lends mathematical rigor to the intuition that agents should 'hedge their bets' in the face of epistemic uncertainty. This property also means that the simple approach of selecting the argmax action is no longer adequate for policy improvement. In Appendix D.2.2 we discuss a policy improvement procedure that takes into account the proximal penalty to find the stochastic optimal policy.

**Implications for RL.** Finally, due to the close connection between the FDPO and RL settings, this work has implications for deep reinforcement learning. Many popular deep RL algorithms utilize a replay buffer to break the correlation between samples in each minibatch (Mnih et al., 2015). However, since these algorithms typically alternate between collecting data and training the network, the replay buffer can also be viewed as a 'temporarily fixed' dataset during the training phase. These algorithms are often very sensitive to hyperparemters; in particular, they perform poorly when the number of learning steps per interaction is large (Fedus et al., 2020). This effect can be explained by our analysis: additional steps of learning cause the policy to approach its naïve FDPO fixed-point, which has poor worst-case suboptimality. A pessimistic algorithm with a better fixed-point could therefore allow us to train more per interaction, improving sample efficiency. A potential direction of future work is therefore to incorporate pessimism into deep RL.

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

## A  BACKGROUND

We write vectors using bold lower-case letters, $\mathbf{a}$, and matrices using upper-case letters, $A$. To refer to individual cells of a vector or rows of a matrix, we use function notation, $\mathbf{a}(x)$. We write the identity matrix as $I$. We use the notation $\mathbb{E}_p[\cdot]$ to denote the average value of a function under a distribution $p$, i.e. for any space $\mathcal{X}$, distribution $p \in Dist(\mathcal{X})$, and function $\mathbf{a} : \mathcal{X} \to \mathbb{R}$, we have $\mathbb{E}_p[\mathbf{a}] := \mathbb{E}_{x \sim p}[\mathbf{a}(x)]$.

When applied to vectors or matrices, we use $<, >, \leq, \geq$ to denote element-wise comparison. Similarly, we use $|\cdot|$ to denote the element-wise absolute value of a vector: $|\mathbf{a}|(x) = |\mathbf{a}(x)|$. We use $|\mathbf{a}|_+$ to denote the element-wise maximum of $\mathbf{a}$ and the zero vector. To denote the total variation distance between two probability distributions, we use $\text{TV}(\mathbf{p}, \mathbf{q}) = \frac{1}{2}|\mathbf{p} - \mathbf{q}|_1$. When $P$ and $Q$ are conditional probability distributions, we adopt the convention $\text{TV}_{\mathcal{X}}(\mathbf{p}, \mathbf{q}) = \langle \frac{1}{2}|\mathbf{p}(\cdot|x) - \mathbf{q}(\cdot|x)|_1 : x \in \mathcal{X}\rangle$, i.e., the vector of total variation distances conditioned on each $x \in \mathcal{X}$.

**Markov Decision Processes.** We represent the environment with which we are interacting as a Markov Decision Process (MDP), defined in standard fashion: $\mathcal{M} := \langle \mathcal{S}, \mathcal{A}, \mathcal{R}, P, \gamma, \rho \rangle$. $\mathcal{S}$ and $\mathcal{A}$ denote the state and action space, which we assume are discrete. We use $\mathcal{Z} := \mathcal{S} \times \mathcal{A}$ as the shorthand for the joint state-action space. The reward function $\mathcal{R} \colon \mathcal{Z} \to Dist([0,1])$ maps state-action pairs to distributions over the unit interval, while the transition function $P \colon \mathcal{Z} \to Dist(\mathcal{S})$ maps state-action pairs to distributions over next states. Finally, $\rho \in Dist(\mathcal{S})$ is the distribution over initial states. We use $\mathbf{r}$ to denote the expected reward function, $\mathbf{r}(\langle s, a \rangle) := \mathbb{E}_{r \sim \mathcal{R}(\cdot|\langle s,a \rangle)}[r]$, which can also be interpreted as a vector $\mathbf{r} \in \mathbb{R}^{|\mathcal{Z}|}$. Similarly, note that $P$ can be described as $P \colon (\mathcal{Z} \times \mathcal{S}) \to \mathbb{R}$, which can be represented as a stochastic matrix $P \in \mathbb{R}^{|\mathcal{Z}| \times |\mathcal{S}|}$. In order to emphasize that these reward and transition functions correspond to the true environment, we sometimes equivalently denote them as $\mathbf{r}_{\mathcal{M}}, P_{\mathcal{M}}$. To denote the vectors of a constant whose sizes are the state and state-action space, we use a single dot to mean state and two dots to mean state-action, e.g., $\dot{\mathbf{1}} \in \mathbb{R}^{|\mathcal{S}|}$ and $\ddot{\mathbf{1}} \in \mathbb{R}^{|\mathcal{Z}|}$. A policy $\pi \colon \mathcal{S} \to Dist(\mathcal{A})$ defines a distribution over actions, conditioned on a state. We denote the space of all possible policies as $\Pi$. We define an "activity matrix" for each policy, $A^{\pi} \in \mathbb{R}^{\mathcal{S} \times \mathcal{Z}}$, which encodes the state-conditional state-action distribution of $\pi$, by letting $A^{\pi}(s, \langle \dot{s}, a \rangle) := \pi(a|s)$ if $s = \dot{s}$, otherwise $A^{\pi}(s, \langle \dot{s}, a \rangle) := 0$. Acting in the MDP according to $\pi$ can thus be represented by $A^{\pi} P \in \mathbb{R}^{|\mathcal{S}| \times |\mathcal{S}|}$ or $P A^{\pi} \in \mathbb{R}^{|\mathcal{Z}| \times |\mathcal{Z}|}$. We define a value function as any $v \colon \Pi \to \mathcal{S} \to \mathbb{R}$ or $q \colon \Pi \to \mathcal{Z} \to \mathbb{R}$ whose output is bounded by $[0, \frac{1}{1-\gamma}]$. Note that this is a slight generalization of the standard definition (Sutton & Barto, 2018) since it accepts a policy as an input. We use the shorthand $\mathbf{v}^{\pi} := \mathbf{v}(\pi)$ and $\mathbf{q}^{\pi} := \mathbf{q}(\pi)$ to denote the result of applying a value function to a specific policy, which can also be represented as a vector, $\mathbf{v}^{\pi} \in \mathbb{R}^{|\mathcal{S}|}$ and $\mathbf{q}^{\pi} \in \mathbb{R}^{|\mathcal{Z}|}$. To denote the output of an arbitrary value function on an arbitrary policy, we use unadorned $\mathbf{v}$ and $\mathbf{q}$. The *expected return* of an MDP $\mathcal{M}$, denoted $\mathbf{v}_{\mathcal{M}}$ or $\mathbf{q}_{\mathcal{M}}$, is the discounted sum of rewards acquired when interacting with the environment:

$$\mathbf{v}_{\mathcal{M}}(\pi) := \sum_{t=0}^{\infty} \left( \gamma A^{\pi} P \right)^t A^{\pi} \mathbf{r} \qquad \qquad \mathbf{q}_{\mathcal{M}}(\pi) := \sum_{t=0}^{\infty} \left( \gamma P A^{\pi} \right)^t \mathbf{r}$$

Note that $\mathbf{v}_{\mathcal{M}}^{\pi} = A^{\pi} \mathbf{q}_{\mathcal{M}}^{\pi}$. An *optimal policy* of an MDP, which we denote $\pi_{\mathcal{M}}^{*}$, is a policy for which the expected return $\mathbf{v}_{\mathcal{M}}$ is maximized under the initial state distribution: $\pi_{\mathcal{M}}^{*} := \arg\max_{\pi} \mathbb{E}_{\rho}[\mathbf{v}_{\mathcal{M}}^{\pi}]$. The statewise expected returns of an optimal policy can be written as $\mathbf{v}_{\mathcal{M}}^{\pi_{\mathcal{M}}^{*}}$. Of particular interest are value functions whose outputs obey fixed-point relationships, $\mathbf{v}^{\pi} = f(\mathbf{v}^{\pi})$ for some $f : (\mathcal{S} \to \mathbb{R}) \to (\mathcal{S} \to \mathbb{R})$. The Bellman consistency equation for a policy $\pi$, $\mathcal{B}_{\mathcal{M}}^{\pi}(\mathbf{x}) := A^{\pi}(\mathbf{r} + \gamma P \mathbf{x})$, which uniquely identifies the vector of expected returns for $\pi$, since $\mathbf{v}_{\mathcal{M}}^{\pi}$ is the only vector for which $\mathbf{v}_{\mathcal{M}}^{\pi} = \mathcal{B}_{\mathcal{M}}^{\pi}(\mathbf{v}_{\mathcal{M}}^{\pi})$ holds. Finally, for any state $s$, the probability of being in the state $s'$ after $t$ time steps when following policy $\pi$ is $[(A^{\pi} P)^t](s, s')$. Furthermore, $\sum_{t=0}^{\infty} (\gamma A^{\pi} P)^t = (I - \gamma A^{\pi} P)^{-1}$. We refer to $(I - \gamma A^{\pi} P)^{-1}$ as the discounted visitation of $\pi$.

**Datasets.** We next introduce basic concepts that are helpful for framing the problem of fixed-dataset policy optimization. We define a *dataset* of $d$ transitions $D := \{\langle s, a, r, s' \rangle\}^d$, and denote the space of all datasets as $\mathcal{D}$. In this work, we specifically consider datasets sampled from a *data distribution* $\Phi : Dist(\mathcal{Z})$; for example, the distribution of state-actions reached by following some stationary policy. We use $D \sim \Phi_d$ to denote constructing a dataset of $d$ tuples $\langle s, a, r, s' \rangle$, by first sampling each $\langle s, a \rangle \sim \Phi$, and then sampling $r$ and $s'$ i.i.d. from the environment reward function and transition function respectively, i.e. each $r \sim \mathcal{R}(\cdot|\langle s, a \rangle)$ and $s' \sim P(\cdot|\langle s, a \rangle)$.[5] We sometimes index $D$ using function notation, using $D(s, a)$ to denote the multiset of all $\langle r, s' \rangle$ such that $\langle s, a, r, s' \rangle \in D$. We use $\ddot{\mathbf{n}}_D \in \mathbb{R}^{|\mathcal{Z}|}$ to denote the vectors of counts, that is, $\ddot{\mathbf{n}}_D(\langle s, a \rangle) := |D(s, a)|$. We sometimes use state-wise versions of these vectors, which we denote with $\dot{\mathbf{n}}_D$. It is further useful to consider the maximum-likelihood reward and transition functions, computed by averaging all rewards and transitions observed in the dataset for each state-action. To this end, we define empirical reward vector $\mathbf{r}_D(\langle s, a \rangle) := \sum_{r,s' \in D(\langle s,a \rangle)} \frac{r}{|D(\langle s,a \rangle)|}$ and empirical transition matrix $P_D(s'|\langle s, a \rangle) := \sum_{r,\dot{s}' \in D(\langle s,a \rangle)} \frac{\mathbb{I}(\dot{s}' = s')}{|D(\langle s,a \rangle)|}$ at all state-actions for which $\ddot{\mathbf{n}}_D(\langle s, a \rangle) > 0$. Where with $\ddot{\mathbf{n}}_D(\langle s, a \rangle) = 0$, there is no clear way to define the maximum-likelihood estimates of

---

[5]Note that this is in some sense a simplifying assumption. In practice, datasets will typically be collected using a trajectory from a non-stationary policy, rather than i.i.d. sampling from the stationary distribution of a stationary policy. This greatly complicates the analysis, so we do not consider that setting in this work.

reward and transition, so we do not specify them. All our results hold no matter how these values are chosen, so long as $\mathbf{r}_D \in [0, \frac{1}{1-\gamma}]$ and $P_D$ is stochastic. The empirical policy of a dataset $D$ is defined as $\hat{\pi}_D(a|s) := \frac{|D(\langle s,a \rangle)|}{|D(\langle s,\cdot \rangle)|}$ except where $\ddot{\mathbf{n}}_D(\langle s, a \rangle) = 0$, where it can similarly be any valid action distribution. The empirical visitation distribution of a dataset $D$ is computed in the same way as the visitation distribution, but with $P_D$ replacing $P$, i.e. $(I - \gamma A^\pi P_D)^{-1}$.

**Problem setting.** The primary focus of this work is on the properties of *fixed-dataset policy optimization (FDPO)* algorithms. These algorithms take the form of a function $\mathcal{O} : \mathcal{D} \to \Pi$, which maps from a dataset to a policy.[6] Note that in this work, we consider $D \sim \Phi_d$, so the dataset is a random variable, and therefore $\mathcal{O}(D)$ is also a random variable. The goal of any FDPO algorithm is to output a policy with minimum suboptimality, i.e. maximum return. Suboptimality is a random variable dependent on $D$, computed by taking the difference between the expected return of an optimal policy and the learned policy under the initial state distribution,

$$\textsc{SubOpt}(\mathcal{O}(D)) = \mathbb{E}_\rho[\mathbf{v}_\mathcal{M}^{\pi_\mathcal{M}^*}] - \mathbb{E}_\rho[\mathbf{v}_\mathcal{M}^{\mathcal{O}(D)}].$$

A related concept is the *fixed-dataset policy evaluation* algorithm, which is any function $\mathcal{E} : \mathcal{D} \to \Pi \to \mathcal{S} \to \mathbb{R}$, which uses a dataset to compute a value function. In this work, we focus our analysis on *value-based FDPO algorithms*, the subset of FDPO algorithms that utilize FDPE algorithms.[7] A value-based FDPO algorithm with FDPE subroutine $\mathcal{E}_{\text{sub}}$ is any algorithm with the following structure:

$$\mathcal{O}_{\text{sub}}^{\text{VB}}(D) := \arg\max_\pi \mathbb{E}_\rho[\mathcal{E}_{\text{sub}}(D, \pi)].$$

We define a *fixed-point family of algorithms*, sometimes referred to as just a *family*, in the following way. Any family called *family* is based on a specific fixed-point identity $f_{\text{family}}$. We use the notation $\mathcal{E}_{\text{family}}$ to denote any FDPE algorithm whose output $\mathbf{v}_D^\pi := \mathcal{E}_{\text{family}}(D, \pi)$ obeys $\mathbf{v}_D^\pi = f_{\text{family}}(\mathbf{v}_D^\pi)$. Finally, $\mathcal{O}_{\text{family}}^{\text{VB}}$ refers to any value-based FDPO algorithm whose subroutine is $\mathcal{E}_{\text{family}}$. We call the set of all algorithms that could implement $\mathcal{E}_{\text{family}}$ the *family of FDPE algorithms*, and the set of all algorithms that could implement $\mathcal{O}_{\text{family}}^{\text{VB}}$ as the *family of FDPO algorithms*.

# B UNCERTAINTY

Epistemic uncertainty measures an agent's knowledge about the world, and is therefore a core concept in reinforcement learning, both for exploration and exploitation; it plays an important role in this work. Most past analyses in the literature implicitly compute some form of epistemic uncertainty to derive their bounds. An important analytical choice in this work is to cleanly separate the estimation of epistemic uncertainty from the problem of decision making. Our approach is to first define a notion of uncertainty as a function with certain properties, then assume that such a function exists, and provide the remainder of our technical results under such an assumption. We also describe several approaches to computing uncertainty.

**Definition 5** (Uncertainty). *A function* $\mathbf{u}_{D,\delta} : \mathcal{Z} \to \mathbb{R}$ *is a* state-action-wise Bellman uncertainty function *if for a dataset* $D \sim \Phi_d$, *it obeys with probability at least* $1 - \delta$ *for all policies* $\pi$ *and all values* $\mathbf{v}$:

$$\mathbf{u}_{D,\delta} \geq |(\mathbf{r}_\mathcal{M} + \gamma P_\mathcal{M} \mathbf{v}) - (\mathbf{r}_D + \gamma P_D \mathbf{v})|$$

---

[6]This formulation hides a dependency on $\rho$, the start-state distribution of the MDP. In general, $\rho$ can be estimated from the dataset $D$, but this estimation introduces some error that affects the analysis. In this work, we assume for analytical simplicity that $\rho$ is known a priori. Technically, this means that it must be provided as an input to $\mathcal{O}$. We hide this dependency for notational clarity.

[7]Intuitively, these algorithms use a policy evaluation subroutine to convert a dataset into a value function, and return an optimal policy according to that value function. Importantly, this definition constrains only the objective of the approach, not its actual algorithmic implementation, i.e., it includes algorithms which never actually invoke the FDPE subroutine. For many online model-free reinforcement learning algorithms, including policy iteration, value iteration, and Q-learning, we can construct closely analogous value-based FDPO algorithms; see Appendix D.1. Furthermore, model-based techniques can be interpreted as using a model to implicitly define a value function, and then optimizing that value function; our results also apply to model-based approaches.

*A function* $\mathbf{u}_{D,\delta}^{\pi} : \mathcal{S} \rightarrow \mathbb{R}$ *is a* state-wise Bellman uncertainty function *if for a dataset* $D \sim \Phi_d$ *and policy* $\pi$, *it obeys with probability at least* $1 - \delta$ *for all policies* $\pi$ *and all values* $\mathbf{v}$:

$$\mathbf{u}_{D,\delta}^{\pi} \geq |A^{\pi}(\mathbf{r}_{\mathcal{M}} + \gamma P_{\mathcal{M}}\mathbf{v}) - A^{\pi}(\mathbf{r}_D + \gamma P_D\mathbf{v})|$$

*A function* $\boldsymbol{\mu}_{D,\delta}^{\pi} : \mathcal{S} \rightarrow \mathbb{R}$ *is a* value uncertainty function *if for a dataset* $D \sim \Phi_d$ *and policy* $\pi$, *it obeys with probability at least* $1 - \delta$ *for all policies* $\pi$ *and all values* $\mathbf{v}$:

$$\boldsymbol{\mu}_{D,\delta}^{\pi} \geq \sum_{t=0}^{\infty} (\gamma A^{\pi} P_D)^t |A^{\pi}(\mathbf{r}_{\mathcal{M}} + \gamma P_{\mathcal{M}}\mathbf{v}) - A^{\pi}(\mathbf{r}_D + \gamma P_D\mathbf{v})|$$

We refer to the quantity returned by an uncertainty function as *uncertainty*, e.g., value uncertainty refers to the quantity returned by a value uncertainty function.

Given a state-action-wise Bellman uncertainty function $\mathbf{u}_{D,\delta}$, it is easy to verify that $A^{\pi}\mathbf{u}_{D,\delta}$ is a state-wise Bellman uncertainty function. Similarly, given a state-wise Bellman uncertainty function $\mathbf{u}_{D,\delta}^{\pi}$, it is easy to verify that $(I - \gamma A^{\pi}P_D)^{-1}\mathbf{u}_{D,\delta}^{\pi}$ is a value uncertainty function. Uncertainty functions which can be constructed in this way are called *decomposable*.

**Definition 6** (Decomposablility)**.** *A state-wise Bellman uncertainty function* $\mathbf{u}_{D,\delta}^{\pi}$ *is state-action-wise decomposable if there exists a state-action-wise Bellman uncertainty function* $\mathbf{u}_{D,\delta}$ *such that* $\mathbf{u}_{D,\delta}^{\pi} = A^{\pi}\mathbf{u}_{D,\delta}$. *A value uncertainty function* $\boldsymbol{\mu}$ *is state-wise decomposable if there exists a state-wise Bellman uncertainty function* $\mathbf{u}_{D,\delta}^{\pi}$ *such that* $\boldsymbol{\mu} = (I - \gamma A^{\pi}P_D)^{-1}\mathbf{u}_{D,\delta}^{\pi}$, *and further, it is state-action-wise decomposable if that* $\mathbf{u}_{D,\delta}^{\pi}$ *is itself state-action-wise decomposable.*

Our definition of Bellman uncertainty captures the intuitive notion that the uncertainty at each state captures how well an approximate Bellman update matches the true environment. Value uncertainty represents the accumulation of these errors over all future timesteps.

**How do these definitions correspond to our intuitions about uncertainty?** An application of the environment's true Bellman update can be viewed as updating the value of each state to reflect information about its future. However, no algorithm in the FDPO setting can apply such an update, because the true environment dynamics are unknown. We may use the dataset to estimate what such an update would look like, but since the limited information in the dataset may not fully specify the properties of the environment, this update will be slightly wrong. It is intuitive to say that the uncertainty at each state corresponds to how well the approximate update matches the truth. Our definition of Bellman uncertainty captures precisely this notion. Further, value uncertainty represents the accumulation of these errors over all future timesteps.

**How can we algorithmically implement uncertainty functions?** In other words, how can we compute a function with the properties required for Definition 5, using only the information in the dataset? All forms of uncertainty are upper-bounds to a quantity, and a tighter upper-bound means that other results down the line which leverage this quantity will be improved. Therefore, it is worth considering this question carefully.

A trivial approach to computing any uncertainty function is simply to return $\frac{1}{1-\gamma}$ for all $s$ and $\langle s, a \rangle$. But although this is technically a valid uncertainty function, it is not very useful, because it does not concentrate with data and does not distinguish between certain and uncertain states. It is very loose and so leads to poor guarantees.

In tabular environments, one way to implement a Bellman uncertainty function is to use a concentration inequality. Depending on which concentration inequality is used, many Bellman uncertainty functions are possible. These approaches lead to Bellman uncertainty which is lower at states with more data, typically in proportion to the square root of the count. To illustrate how to do this, we show in Appendix B.1 how a basic application of Hoeffding's inequality can be used to derive a state-action-wise Bellman uncertainty. In Appendix B.2, we show an alternative application of Hoeffding's which results in a state-wise Bellman uncertainty, which is a tighter bound on error. In Appendix B.3, we discuss other techniques which may be useful in tightening further.

When the value function is represented by a neural network, it is not currently known how to implement a Bellman uncertainty function. When an empirical Bellman update is applied to a neural

network, the change in value of any given state is impacted by generalization from other states. Therefore, the counts are not meaningful, and concentration inequalities are not applicable. In the neural network literature, many "uncertainty estimation techniques" have been proposed, which capture something analogous to an intuitive notion of uncertainty; however, none are principled enough to be useful in computing Bellman uncertainty.

## B.1 STATE-ACTION-WISE BOUND

We seek to construct an $\mathbf{u}_{D,\delta}^{\pi}$ such that $|A^{\pi}(\mathbf{r}_{\mathcal{M}} + \gamma P_{\mathcal{M}}\mathbf{v}) - A^{\pi}(\mathbf{r}_D + \gamma P_D\mathbf{v})| \leq \mathbf{u}_{D,\delta}^{\pi}$ with probability at least $1 - \delta$.

Firstly, let's consider the simplest possible bound. $\mathbf{v}$ is bounded in $[0, \frac{1}{1-\gamma}]$, so both $A^{\pi}(\mathbf{r}_{\mathcal{M}} + \gamma P_{\mathcal{M}}\mathbf{v})$ and $A^{\pi}(\mathbf{r}_D + \gamma P_D\mathbf{v})$ must be as well. Thus, their difference is also bounded:

$$|A^{\pi}(\mathbf{r}_{\mathcal{M}} + \gamma P_{\mathcal{M}}\mathbf{v}) - A^{\pi}(\mathbf{r}_D + \gamma P_D\mathbf{v})| \leq \frac{1}{1-\gamma}$$

Next, consider that for any $\langle s, a \rangle$, the expression $\mathbf{r}_D(\langle s, a \rangle) + \gamma P_D(\langle s, a \rangle)\mathbf{v}^{\pi}$ can be equivalently expressed as an expectation of random variables,

$$\mathbf{r}_D(\langle s, a \rangle) + \gamma P_D(\langle s, a \rangle)\mathbf{v} = \frac{1}{\ddot{\mathbf{n}}_D(\langle s, a \rangle)} \sum_{r,s' \in D(\langle s,a \rangle)} r + \gamma\mathbf{v}(s'),$$

each with expected value

$$\mathbb{E}_{r,s' \in D(\langle s,a \rangle)}[r + \gamma\mathbf{v}(s')] = \mathbb{E}_{\substack{r \sim \mathcal{R}(\cdot|\langle s,a \rangle) \\ s' \sim P(\cdot|\langle s,a \rangle)}}[r + \gamma\mathbf{v}(s')] = [\mathbf{r}_{\mathcal{M}} + \gamma P_{\mathcal{M}}\mathbf{v}](\langle s, a \rangle).$$

Note also that each of these random variables is bounded $[0, \frac{1}{1-\gamma}]$. Thus, Hoeffding's inequality tells us that this mean of bounded random variables must be close to their expectation with high probability. By invoking Hoeffding's at each of the $|\mathcal{S} \times \mathcal{A}|$ state-actions, and taking a union bound, we see that with probability at least $1 - \delta$,

$$|(\mathbf{r}_{\mathcal{M}} + \gamma P_{\mathcal{M}}\mathbf{v}) - (\mathbf{r}_D + \gamma P_D\mathbf{v})| \leq \frac{1}{1-\gamma}\sqrt{\frac{1}{2}\ln\frac{2|\mathcal{S} \times \mathcal{A}|}{\delta}}\ddot{\mathbf{n}}_D^{-1}$$

We can left-multiply $A^{\pi}$ and rearrange to get:

$$|A^{\pi}(\mathbf{r}_{\mathcal{M}} + \gamma P_{\mathcal{M}}\mathbf{v}) - A^{\pi}(\mathbf{r}_D + \gamma P_D\mathbf{v}_{\mathcal{M}}^{\pi})| \leq \left(\frac{1}{1-\gamma}\sqrt{\frac{1}{2}\ln\frac{2|\mathcal{S} \times \mathcal{A}|}{\delta}}\right)A^{\pi}\ddot{\mathbf{n}}_D^{-\frac{1}{2}}$$

Finally, we simply intersect this bound with the $\frac{1}{1-\gamma}$ bound from earlier. Thus, we see that with probability at least $1 - \delta$,

$$|A^{\pi}(\mathbf{r}_{\mathcal{M}} + \gamma P_{\mathcal{M}}\mathbf{v}) - A^{\pi}(\mathbf{r}_D + \gamma P_D\mathbf{v}_{\mathcal{M}}^{\pi})| \leq \frac{1}{1-\gamma} \cdot \min\left(\left(\sqrt{\frac{1}{2}\ln\frac{2|\mathcal{S} \times \mathcal{A}|}{\delta}}\right)A^{\pi}\ddot{\mathbf{n}}_D^{-\frac{1}{2}}, 1\right)$$

## B.2 STATE-WISE BOUND

This bound is similar to the previous, but uses Hoeffding's to bound the value at each state all at once, rather than bounding the value at each state-action.

Choose a collection of possible state-local policies $\Pi_{\text{local}} \subseteq Dist(\mathcal{A})$. Each state-local policy is a member of the action simplex.

For any $s \in \mathcal{S}$ and $\pi \in \Pi_{\text{local}}$, the expression $[A^{\pi}(\mathbf{r}_D + \gamma P_D\mathbf{v})](s)$ can be equivalently expressed as a mean of random variables,

$$[A^{\pi}(\mathbf{r}_D + \gamma P_D\mathbf{v})](s) = \frac{1}{\dot{\mathbf{n}}_D(s)} \sum_{a,r,s' \in D(s)} \frac{\pi(a|s)}{\hat{\pi}_D(a|s)}(r + \gamma\mathbf{v}(s')),$$

each with expected value

$$\mathbb{E}_{a,r,s' \in D(s)} \left[ \frac{\pi(a|s)}{\hat{\pi}_D(a|s)} (r + \gamma \mathbf{v}(s')) \right] = \mathbb{E}_{\substack{a \sim \pi(\cdot|s) \\ r \sim \mathcal{R}(\cdot|\langle s,a \rangle) \\ s' \sim P(\cdot|\langle s,a \rangle)}} [r + \gamma \mathbf{v}(s')] = [A^\pi (\mathbf{r}_\mathcal{M} + \gamma P_\mathcal{M} \mathbf{v})](s).$$

Note also that each of these random variables is bounded $[0, \frac{1}{1-\gamma} \frac{\pi(a|s)}{\hat{\pi}_D(a|s)}]$. Thus, Hoeffding's inequality tells us that this sum of bounded random variables must be close to its expectation with high probability. By invoking Hoeffding's at each of the $|\mathcal{S}|$ states and $|\Pi_{\text{local}}|$ local policies, and taking a union bound, we see that with probability at least $1 - \delta$,

$$|A^\pi (\mathbf{r}_\mathcal{M} + \gamma P_\mathcal{M} \mathbf{v}) - A^\pi (\mathbf{r}_D + \gamma P_D \mathbf{v}_\mathcal{M}^\pi)| \leq \frac{1}{1-\gamma} \sqrt{\frac{1}{2} \ln \frac{2|\mathcal{S} \times \Pi_{\text{local}}|}{\delta} (A^\pi)^{\circ 2} \ddot{\mathbf{n}}_D^{-1}}$$

where the term $(A^\pi)^{\circ 2}$ refers to the elementwise square. Finally, we once again intersect with $\frac{1}{1-\gamma}$, yielding that with probability at least $1 - \delta$,

$$|A^\pi (\mathbf{r}_\mathcal{M} + \gamma P_\mathcal{M} \mathbf{v}) - A^\pi (\mathbf{r}_D + \gamma P_D \mathbf{v}_\mathcal{M}^\pi)| \leq \frac{1}{1-\gamma} \cdot \min \left( \sqrt{\frac{1}{2} \ln \frac{2|\mathcal{S} \times \Pi_{\text{local}}|}{\delta} (A^\pi)^{\circ 2} \ddot{\mathbf{n}}_D^{-1}}, 1 \right)$$

Comparing this to the Bellman uncertainty function in Appendix B.1, we see two differences. Firstly, we have replaced a factor of $|\mathcal{A}|$ with $|\Pi_{\text{local}}|$, typically loosening the bound somewhat (depending on choice of considered local policies). Secondly, $A^\pi$ has now moved inside of the square root; since the square root is concave, Jensen's inequality says that

$$\sqrt{(A^\pi)^{\circ 2} \ddot{\mathbf{n}}_D^{-1}} \leq \sqrt{(A^\pi)^{\circ 2}} \sqrt{\ddot{\mathbf{n}}_D^{-1}} = A^\pi \ddot{\mathbf{n}}_D^{-\frac{1}{2}}$$

and so this represents a tightening of the bound.

When $\Pi_{\text{local}}$ is the set of deterministic policies, this bound is equivalent to that of Appendix B.1. This can easily be seen by noting that for a deterministic policy, all elements of $(A^\pi)^{\circ 2}$ are either 1 or 0, and so

$$\sqrt{(A^\pi)^{\circ 2} \ddot{\mathbf{n}}_D^{-1}} = A^\pi \ddot{\mathbf{n}}_D^{-\frac{1}{2}}$$

and also that the size of the set of deterministic policies is exactly $|\mathcal{A}|$.

An important property of this bound is that it shows that stochastic policies can be often be evaluated with lower error than deterministic policies. We prove this by example. Consider an MDP with a single state $s$ and two actions $a_0, a_1$, and a dataset with $\ddot{\mathbf{n}}_D(\langle s, a_0 \rangle) = \ddot{\mathbf{n}}_D(\langle s, a_1 \rangle) = 2$. We can parameterize the policy by a single number $\xi \in [0, 1]$ by setting $\pi(a_0|s) = \xi, \pi(a_1|s) = 1 - \xi$. The size of this bound will be proportional to $\sqrt{\frac{\xi^2}{2} + \frac{(1-\xi)^2}{2}}$, and setting the derivative equal to zero, we see that the minimum is $\xi = \frac{1}{2}$. (Of course, we would need to increase the size of our local policy set to include this, in order to be able to actually select it, and doing so will increase the overall bound; this example only shows that for a *given* policy set, the selected policy may in general be stochastic.)

Finding the optimum for larger local policy sets is non-trivial, so we leave a full treatment of algorithms which leverage this bound for future work.

## B.3 OTHER BOUNDS

There are a few other paths by which this bound can be made tighter still. The above bounds take an extra factor of $\frac{1}{1-\gamma}$ because we bound the overall return, rather than simply bounding the reward and transition functions. This was done because a bound on the transition function would add a cost of $O(\sqrt{\mathcal{S}})$. However, this can be mitigated by intersecting the above confidence interval with a Good-Turing interval, as proposed in Taleghan et al. (2015). Doing so will cause the bound to concentrate much more quickly in MDPs where the transition function is relatively deterministic. We expect this to be the case for most practical MDPs.

Similarly, empirical Bernstein confidence intervals can be used in place of Hoeffding's, to increase the rate of concentration for low-variance rewards and transitions (Maurer & Pontil, 2009), leading to improved performance in MDPs where these are common.

Finally, we may be able to apply a concentration inequality in a more advanced fashion to compute a value uncertainty function which is *not* statewise decomposable: we bound some useful notion of value error directly, rather than computing the Bellman uncertainty function and taking the visitation-weighted sum. This would result in an overall tighter bound on value uncertainty by hedging over data across multiple timesteps. However, in doing so, we would sacrifice the monotonic improvement property needed for convergence of algorithms like policy iteration. This idea has a parallel in the robust MDP literature. The bounds in Appendix B.1 can be seen as constructing an *sa-rectangular* robust MDP, whereas Appendix B.2 is similar to constructing an *s-rectangular* robust MDP Wiesemann et al. (2013). More recently, approaches have been proposed which go beyond s-rectangular (Goyal & Grand-Clement, 2018), and such approaches likely have natural parallels in implementing value uncertainty functions.

# C  PROOFS

## C.1  PROOF OF OVER/UNDER DECOMPOSITION

Starting from the definition of suboptimality, we see

$$
\begin{aligned}
\text{SUBOPT}(\mathcal{O}^{\text{VB}}(D)) &= \mathbb{E}_\rho[\mathbf{v}_{\mathcal{M}}^{\pi_{\mathcal{M}}^*}] - \mathbb{E}_\rho[\mathbf{v}_{\mathcal{M}}^{\pi_D^*}] \\
&= \mathbb{E}_\rho[\mathbf{v}_{\mathcal{M}}^{\pi_{\mathcal{M}}^*} + (-\mathbf{v}_D^\pi + \mathbf{v}_D^\pi) + (-\mathbf{v}_D^{\pi_D^*} + \mathbf{v}_D^{\pi_D^*}) - \mathbf{v}_{\mathcal{M}}^{\pi_D^*}] && \text{(valid for any } \pi) \\
&\leq \mathbb{E}_\rho[\mathbf{v}_{\mathcal{M}}^* - \mathbf{v}_D^\pi] + \mathbb{E}_\rho[\mathbf{v}_D^{\pi_D^*} - \mathbf{v}_{\mathcal{M}}^{\pi_D^*}] && \text{(using } \mathbb{E}_\rho[\mathbf{v}_D^\pi - \mathbf{v}_D^{\pi_D^*}] \leq 0)
\end{aligned}
$$

Since the above holds for all $\pi$,

$$
\begin{aligned}
\text{SUBOPT}(\mathcal{O}^{\text{VB}}(D)) &\leq \inf_\pi \left( \mathbb{E}_\rho[\mathbf{v}_{\mathcal{M}}^{\pi_{\mathcal{M}}^*} - \mathbf{v}_D^\pi] \right) + \mathbb{E}_\rho[\mathbf{v}_D^{\pi_D^*} - \mathbf{v}_{\mathcal{M}}^{\pi_D^*}] \\
&\leq \inf_\pi \left( \mathbb{E}_\rho[\mathbf{v}_{\mathcal{M}}^{\pi_{\mathcal{M}}^*} - \mathbf{v}_D^\pi] \right) + \sup_\pi \left( \mathbb{E}_\rho[\mathbf{v}_D^\pi - \mathbf{v}_{\mathcal{M}}^\pi] \right) && \text{(using } \mathbf{v}_D^{\pi_D^*} \in \Pi) \\
&= \inf_\pi \left( \mathbb{E}_\rho[\mathbf{v}_{\mathcal{M}}^{\pi_{\mathcal{M}}^*} - \mathbf{v}_{\mathcal{M}}^\pi] + \mathbb{E}_\rho[\mathbf{v}_{\mathcal{M}}^\pi - \mathbf{v}_D^\pi] \right) + \sup_\pi \left( \mathbb{E}_\rho[\mathbf{v}_D^\pi - \mathbf{v}_{\mathcal{M}}^\pi] \right)
\end{aligned}
$$

## C.2  TIGHTNESS OF OVER/UNDER DECOMPOSITION

We show that the bound given in Lemma 1 is tight via a simple example.

*Proof.* Consider a bandit-structured MDP with a single state and two actions, A and B, with rewards of 0 and 1 respectively, which both lead to terminal states.

First, consider the left-hand side. If an FDPE subroutine estimates the value of both arms to be 1, then the policy which always selects arm A is an optimal policy of the corresponding FDPO algorithm. In this case, the suboptimality is clearly equal to 1. This is clearly the worst-case suboptimality, since it is the largest possible suboptimality in the environment.

On the right-hand side, note that term (A) is 0 when $\pi$ is the policy that always picks B, while term (B) is 1 when $\pi$ is the policy that always picks A. Thus, the left-hand and right-hand sides are equal, and the bound is tight. □

## C.3  RESIDUAL VISITATION LEMMA

We prove a basic lemma showing that the error of any value function is equal to its one-step Bellman residual, summed over its visitation distribution. Though this result is well-known, it is not clearly stated elsewhere in the literature, so we prove it here for clarity.

**Lemma 2.** *For any MDP $\xi$ and policy $\pi$, consider the Bellman fixed-point equation given by, let $\mathbf{v}_\xi^\pi$ be defined as the unique value vector such that $\mathbf{v}_\xi^\pi = A^\pi(\mathbf{r}_\xi + \gamma P_\xi \mathbf{v}_\xi^\pi)$, and let $\mathbf{v}$ be any other value*

*vector. We have*

$$\mathbf{v}_\xi^\pi - \mathbf{v} = (I - \gamma A^\pi P_\xi)^{-1}(A^\pi(\mathbf{r}_\xi + \gamma P_\xi \mathbf{v}) - \mathbf{v}) \tag{1}$$

$$\mathbf{v} - \mathbf{v}_\xi^\pi = (I - \gamma A^\pi P_\xi)^{-1}(\mathbf{v} - A^\pi(\mathbf{r}_\xi + \gamma P_\xi \mathbf{v})) \tag{2}$$

$$|\mathbf{v}_\xi^\pi - \mathbf{v}| = (I - \gamma A^\pi P_\xi)^{-1}|A^\pi(\mathbf{r}_\xi + \gamma P_\xi \mathbf{v}) - \mathbf{v}| \tag{3}$$

*Proof.*

$$
\begin{aligned}
A^\pi(\mathbf{r}_\xi + \gamma P_\xi \mathbf{v}) - \mathbf{v} &= A^\pi(\mathbf{r}_\xi + \gamma P_\xi \mathbf{v}) - \mathbf{v}_\xi^\pi + \mathbf{v}_\xi^\pi - \mathbf{v} \\
&= A^\pi(\mathbf{r}_\xi + \gamma P_\xi \mathbf{v}) - A^\pi(\mathbf{r}_\xi + \gamma P_\xi \mathbf{v}_\xi^\pi) + \mathbf{v}_\xi^\pi - \mathbf{v} \\
&= \gamma A^\pi P_\xi(\mathbf{v} - \mathbf{v}_\xi^\pi) + (\mathbf{v}_\xi^\pi - \mathbf{v}) \\
&= (\mathbf{v}_\xi^\pi - \mathbf{v}) - \gamma A^\pi P_\xi(\mathbf{v}_\xi^\pi - \mathbf{v}) \\
&= (I - \gamma A^\pi P_\xi)(\mathbf{v}_\xi^\pi - \mathbf{v})
\end{aligned}
$$

Thus, we see $(I - \gamma A^\pi P_\xi)^{-1}(A^\pi(\mathbf{r}_\xi + \gamma P_\xi \mathbf{v}) - \mathbf{v}) = \mathbf{v}_\xi^\pi - \mathbf{v}$. An identical proof can be completed starting with $\mathbf{v} - A^\pi(\mathbf{r}_\xi + \gamma P_\xi \mathbf{v})$, leading to the desired result. $\square$

## C.4  NAÏVE FDPE ERROR BOUND

We show how the error of naïve FDPE algorithms is bounded by the value uncertainty of state-actions visited by the policy under evaluation. Next, in Section 4, we use this bound to derive a suboptimality guarantee for naïve FDPO.

**Lemma 3** (Naïve FDPE error bound). *Consider any naïve fixed-dataset policy evaluation algorithm $\mathcal{E}_{naïve}$. For any policy $\pi$ and dataset $D$, denote $\mathbf{v}_D^\pi := \mathcal{E}_{naïve}(D, \pi)$. Let $\boldsymbol{\mu}_{D,\delta}^\pi$ be any value uncertainty function. The following component-wise bound holds with probability at least $1 - \delta$:*

$$|\mathbf{v}_\mathcal{M}^\pi - \mathbf{v}_D^\pi| \leq \boldsymbol{\mu}_{D,\delta}^\pi$$

*Proof.* Notice that the naïve fixed-point function is equivalent to the Bellman fixed-point equation for a specific MDP: the empirical MDP defined by $\langle \mathcal{S}, \mathcal{A}, \mathbf{r}_D, P_D, \gamma, \rho \rangle$. Thus, invoking Lemma 2, for any values $\mathbf{v}$ we have

$$|\mathbf{v}_D^\pi - \mathbf{v}| = (I - \gamma A^\pi P_D)^{-1}|A^\pi(\mathbf{r}_D + \gamma P_D \mathbf{v}) - \mathbf{v}|.$$

Since $\mathbf{v}_\mathcal{M}^\pi$ is a value vector, this immediately implies that

$$|\mathbf{v}_D^\pi - \mathbf{v}_\mathcal{M}^\pi| = (I - \gamma A^\pi P_D)^{-1}|A^\pi(\mathbf{r}_D + \gamma P_D \mathbf{v}_\mathcal{M}^\pi) - \mathbf{v}_\mathcal{M}^\pi|.$$

Since $\mathbf{v}_\mathcal{M}^\pi$ is the solution to the Bellman consistency fixed-point,

$$|\mathbf{v}_D^\pi - \mathbf{v}_\mathcal{M}^\pi| = (I - \gamma A^\pi P_D)^{-1}|A^\pi(\mathbf{r}_D + \gamma P_D \mathbf{v}_\mathcal{M}^\pi) - A^\pi(\mathbf{r}_\mathcal{M} + \gamma P_\mathcal{M} \mathbf{v}_\mathcal{M}^\pi)|.$$

Using the definition of a value uncertainty function $\boldsymbol{\mu}_{D,\delta}^\pi$, we arrive at

$$|\mathbf{v}_D^\pi - \mathbf{v}_\mathcal{M}^\pi| \leq \boldsymbol{\mu}_{D,\delta}^\pi$$

completing the proof. $\square$

Thus, reducing value uncertainty improves our guarantees on evaluation error. For any fixed policy, value uncertainty can be reduced by reducing the Bellman uncertainty on states visited by that policy. In the tabular setting this means observing more interactions from the state-actions that that policy visits frequently. Conversely, for any fixed dataset, we will have a certain Bellman uncertainty in each state, and policies mostly visit low-Bellman-uncertainty states can be evaluated with lower error.[8]

---

[8] In the function approximation setting, we do not necessarily need to observe an interaction with a particular state-action to reduce our Bellman uncertainty on it. This is because observing other state-actions may allow us to reduce Bellman uncertainty through generalization. Similarly, the most-certain policy for a fixed dataset may not be the policy for which we have the most data, but rather, the policy which our dataset *informs* us about the most.

Our bound differs from prior work (Jiang, 2019a; Ghavamzadeh et al., 2016) in that it is significantly more fine-grained. We provide a component-wise bound on error, whereas previous results bound the $l_\infty$ norm. Furthermore, our bounds are sensitive to the Bellman uncertainty in each individual reward and transition, rather than only relying on the most-uncertain. As a result, our bound does not require all states to have the same number of samples, and is non-vacuous even in the case where some state-actions have no data.

Our bound can also be viewed as an extension of work on approximate dynamic programming. In that setting, the literature contains fine-grained results on the accumulation of local errors (Munos, 2007). However, those results are typically understood as applying to errors induced by approximation via some limited function class. Our bound can be seen as an application of those ideas to the case where errors are induced by limited observations.

## C.5 RELATIVE VALUE UNCERTAINTY

The key to the construction of proximal pessimistic algorithms is the relationship between the value uncertainties of any two policies $\pi, \pi'$, for any state-action-wise decomposable value function.

**Lemma 4** (Relative value uncertainty). *For any two policies $\pi, \pi'$, and any state-action-wise decomposable value uncertainty $\boldsymbol{\mu}$, we have*

$$\boldsymbol{\mu}_{D,\delta}^\pi - \boldsymbol{\mu}_{D,\delta}^{\pi'} = (I - \gamma A^\pi P_D)^{-1} \left( (A^\pi - A^{\pi'})(\mathbf{u}_{D,\delta} + \gamma P_D \boldsymbol{\mu}_{D,\delta}^{\pi'}) \right)$$

*Proof.* Firstly, note that since the value uncertainty function is state-action-wise decomposable, we can express it in a Bellman-like form for some state-wise Bellman uncertainty $\mathbf{u}_{D,\delta}^\pi$ as $\boldsymbol{\mu}_{D,\delta}^\pi = (I - \gamma A^\pi P_D)^{-1} \mathbf{u}_{D,\delta}^\pi$. Further, $\mathbf{u}_{D,\delta}^\pi$ is itself state-action-wise decomposable, so it can be written as $\mathbf{u}_{D,\delta}^\pi = A^\pi \mathbf{u}_{D,\delta}$.

We can use this to derive the following relationship.

$$\boldsymbol{\mu}_{D,\delta}^\pi = (I - \gamma A^\pi P_D)^{-1} \mathbf{u}_{D,\delta}^\pi$$
$$\boldsymbol{\mu}_{D,\delta}^\pi - \gamma P_D A^\pi \boldsymbol{\mu}_{D,\delta}^\pi = \mathbf{u}_{D,\delta}^\pi$$
$$\boldsymbol{\mu}_{D,\delta}^\pi = \mathbf{u}_{D,\delta}^\pi + \gamma A^\pi P_D \boldsymbol{\mu}_{D,\delta}^\pi$$

Next, we bound the difference between the value uncertainty of $\pi$ and $\pi'$.

$$\begin{aligned}
\boldsymbol{\mu}_{D,\delta}^\pi - \boldsymbol{\mu}_{D,\delta}^{\pi'} &= (\mathbf{u}_{D,\delta}^\pi + \gamma A^\pi P_D \boldsymbol{\mu}_{D,\delta}^\pi) - (\mathbf{u}_{D,\delta}^{\pi'} + \gamma A^{\pi'} P_D \boldsymbol{\mu}_{D,\delta}^{\pi'}) \\
&= (\mathbf{u}_{D,\delta}^\pi - \mathbf{u}_{D,\delta}^{\pi'}) + \gamma A^\pi P_D \boldsymbol{\mu}_{D,\delta}^\pi - \gamma A^{\pi'} P_D \boldsymbol{\mu}_{D,\delta}^{\pi'} \\
&= (A^\pi - A^{\pi'})\mathbf{u}_{D,\delta} + \gamma A^\pi P_D \boldsymbol{\mu}_{D,\delta}^\pi - \gamma(A^{\pi'} - A^\pi + A^\pi)P_D\boldsymbol{\mu}_{D,\delta}^{\pi'} \\
&= \gamma A^\pi P_D \left( \boldsymbol{\mu}_{D,\delta}^\pi - \boldsymbol{\mu}_{D,\delta}^{\pi'} \right) + (A^\pi - A^{\pi'})\mathbf{u}_{D,\delta} + \gamma(A^\pi - A^{\pi'})P_D\boldsymbol{\mu}_{D,\delta}^{\pi'} \\
&= \gamma A^\pi P_D \left( \boldsymbol{\mu}_{D,\delta}^\pi - \boldsymbol{\mu}_{D,\delta}^{\pi'} \right) + (A^\pi - A^{\pi'})(\mathbf{u}_{D,\delta} + \gamma P_D\boldsymbol{\mu}_{D,\delta}^{\pi'})
\end{aligned}$$

This is a geometric series, so $(I - \gamma A^\pi P_D) \left( \boldsymbol{\mu}_{D,\delta}^\pi - \boldsymbol{\mu}_{D,\delta}^{\pi'} \right) = (A^\pi - A^{\pi'})(\mathbf{u}_{D,\delta} + \gamma P_D\boldsymbol{\mu}_{D,\delta}^{\pi'})$. Left-multiplying by $(I - \gamma A^\pi P_D)^{-1}$ we arrive at the desired result. $\qquad\square$

## C.6 NAÏVE FDPE RELATIVE ERROR BOUND

**Lemma 5** (Naïve FDPE relative error bound). *Consider any naïve fixed-dataset policy evaluation algorithm $\mathcal{E}_{naïve}$. For any policy $\pi$ and dataset $D$, denote $\mathbf{v}_D^\pi := \mathcal{E}_{naïve}(D, \pi)$. Then, for any other policy $\pi'$, the following bound holds with probability at least $1 - \delta$:*

$$|\mathbf{v}_D^\pi - \mathbf{v}_{\mathcal{M}}^\pi| \leq \boldsymbol{\mu}_{D,\delta}^\pi \leq \boldsymbol{\mu}_{D,\delta}^{\pi'} + (I - \gamma A^\pi P_D)^{-1} \left( \frac{1}{(1-\gamma)^2} \right) TV_{\mathcal{S}}(\pi, \pi')$$

*Proof.* The goal of this section is to construct an error bound for which we can optimize $\pi$ without needing to compute any uncertainties. To do this, we must replace this quantity with a looser upper bound.

First, consider a state-action-wise decomposable value uncertainty function $\boldsymbol{\mu}_{D,\delta}^{\pi}$. We have $\boldsymbol{\mu}_{D,\delta}^{\pi} = (I - \gamma P_D A^{\pi})^{-1} \mathbf{u}_{D,\delta}^{\pi}$ for some $\mathbf{u}_{D,\delta}^{\pi}$, where $\mathbf{u}_{D,\delta}^{\pi} = A^{\pi} \mathbf{u}_{D,\delta}$ for some $\mathbf{u}_{D,\delta}$.

Note that state-action-wise Bellman uncertainty can be trivially bounded as $\mathbf{u}_{D,\delta} \leq \frac{1}{1-\gamma} \ddot{\mathbf{i}}$. Also, since $(I - \gamma P_D A^{\pi})^{-1} \leq \frac{1}{1-\gamma}$, any state-action-wise decomposable value uncertainty can be trivially bounded as $\boldsymbol{\mu}_{D,\delta}^{\pi} \leq \frac{1}{(1-\gamma)^2} \ddot{\mathbf{i}}$.

We now invoke Lemma 4. We then substitute the above bounds into the second term, after ensuring that all coefficients are positive.

$$
\begin{aligned}
\boldsymbol{\mu}_{D,\delta}^{\pi} - \boldsymbol{\mu}_{D,\delta}^{\pi'} &= (I - \gamma A^{\pi} P_D)^{-1} \left( (A^{\pi} - A^{\pi'})(\mathbf{u}_{D,\delta} + \gamma P_D \boldsymbol{\mu}_{D,\delta}^{\pi'}) \right) \\
&\leq (I - \gamma A^{\pi} P_D)^{-1} \left( |A^{\pi} - A^{\pi'}|_+ (\mathbf{u}_{D,\delta} + \gamma P_D \boldsymbol{\mu}_{D,\delta}^{\pi'}) \right) \\
&\leq (I - \gamma A^{\pi} P_D)^{-1} \left( |A^{\pi} - A^{\pi'}|_+ \left( \frac{1}{1-\gamma} \ddot{\mathbf{i}} + \gamma P_D \frac{1}{(1-\gamma)^2} \ddot{\mathbf{i}} \right) \right) \\
&= (I - \gamma A^{\pi} P_D)^{-1} \left( |A^{\pi} - A^{\pi'}|_+ \left( \frac{1}{1-\gamma} \ddot{\mathbf{i}} + \frac{\gamma}{(1-\gamma)^2} \ddot{\mathbf{i}} \right) \right) \\
&= (I - \gamma A^{\pi} P_D)^{-1} \left( \frac{1}{(1-\gamma)^2} \right) |A^{\pi} - A^{\pi'}|_+ \ddot{\mathbf{i}} \\
&= (I - \gamma A^{\pi} P_D)^{-1} \left( \frac{1}{(1-\gamma)^2} \right) \mathrm{TV}_{\mathcal{S}}(\pi, \pi')
\end{aligned}
$$

The third-to-last step follows from the fact that $P_D$ is stochastic. The final step follows from the fact that the positive and negative components of the state-wise difference between policies must be symmetric, so $|A^{\pi} - A^{\pi'}|_+ \ddot{\mathbf{i}}$ is precisely equivalent to the state-wise total variation distance, $\mathrm{TV}_{\mathcal{S}}(\pi, \pi')$. Thus, we have

$$
\boldsymbol{\mu}_{D,\delta}^{\pi} \leq \boldsymbol{\mu}_{D,\delta}^{\pi'} + (I - \gamma A^{\pi} P_D)^{-1} \left( \frac{1}{(1-\gamma)^2} \right) \mathrm{TV}_{\mathcal{S}}(\pi, \pi').
$$

Finally, invoking Lemma 3, we arrive at the desired result. $\square$

## C.7 SUBOPTIMALITY OF UNCERTAINTY-AWARE PESSIMISTIC FDPO ALGORITHMS

Let $\mathbf{v}_D^{\pi} := \underline{\mathcal{E}}_{\mathrm{ua}}(D, \pi)$. From the definition of the UA family, we have the fixed-point property $\mathbf{v}_D^{\pi} = A^{\pi}(\mathbf{r}_D + \gamma P_D \mathbf{v}_D^{\pi}) - \alpha \mathbf{u}_{D,\delta}^{\pi}$, and the standard geometric series rearrangement yields $\mathbf{v}_D^{\pi} = (I - \gamma A^{\pi} P_D)^{-1} (A^{\pi} \mathbf{r}_D - \alpha \mathbf{u}_{D,\delta}^{\pi})$. From here, we see:

$$
\begin{aligned}
\mathbf{v}_D^{\pi} &= (I - \gamma A^{\pi} P_D)^{-1} (A^{\pi} \mathbf{r}_D - \alpha \mathbf{u}_{D,\delta}^{\pi}) \\
&= (I - \gamma A^{\pi} P_D)^{-1} A^{\pi} \mathbf{r}_D - (I - \gamma A^{\pi} P_D)^{-1} \alpha \mathbf{u}_{D,\delta}^{\pi} \\
&= \mathcal{E}_{\mathrm{naïve}}(D, \pi) - \alpha \boldsymbol{\mu}_{D,\delta}^{\pi}
\end{aligned}
$$

We now use this to bound overestimation and underestimation error of $\underline{\mathcal{E}}_{\mathrm{ua}}(D, \pi)$ by invoking Lemma 3, which holds with probability at least $1 - \delta$. First, for underestimation, we see:

$$
\begin{aligned}
\mathbf{v}_{\mathcal{M}}^{\pi} - \mathbf{v}_D^{\pi} &= \mathbf{v}_{\mathcal{M}}^{\pi} - \left( \mathcal{E}_{\mathrm{naïve}}(D, \pi) - \alpha \boldsymbol{\mu}_{D,\delta}^{\pi} \right) \\
&= (\mathbf{v}_{\mathcal{M}}^{\pi} - \mathcal{E}_{\mathrm{naïve}}(D, \pi)) + \alpha \boldsymbol{\mu}_{D,\delta}^{\pi} \\
&\leq \boldsymbol{\mu}_{D,\delta}^{\pi} + \alpha \boldsymbol{\mu}_{D,\delta}^{\pi} \\
&= (1 + \alpha) \boldsymbol{\mu}_{D,\delta}^{\pi}
\end{aligned}
$$

and thus, $\mathbf{v}_{\mathcal{M}}^{\pi} - \mathbf{v}_D^{\pi} \leq (1 + \alpha) \boldsymbol{\mu}_{D,\delta}^{\pi}$. Next, for overestimation, we see:

$$
\mathbf{v}_D^{\pi} - \mathbf{v}_{\mathcal{M}}^{\pi} = \left( \mathcal{E}_{\mathrm{naïve}}(D, \pi) - \alpha \boldsymbol{\mu}_{D,\delta}^{\pi} \right) - \mathbf{v}_{\mathcal{M}}^{\pi}
$$

$$
\begin{aligned}
&= (\mathcal{E}_{\text{naïve}}(D, \pi) - \mathbf{v}_{\mathcal{M}}^{\pi}) - \alpha \boldsymbol{\mu}_{D,\delta}^{\pi} \\
&\leq \boldsymbol{\mu}_{D,\delta}^{\pi} - \alpha \boldsymbol{\mu}_{D,\delta}^{\pi} \\
&= (1 - \alpha) \boldsymbol{\mu}_{D,\delta}^{\pi}
\end{aligned}
$$

and thus, $\mathbf{v}_D^{\pi} - \mathbf{v}_{\mathcal{M}}^{\pi} \leq (1 - \alpha) \boldsymbol{\mu}_{D,\delta}^{\pi}$. Substituting these bounds into Lemma 1 gives the desired result.

## C.8 SUBOPTIMALITY OF PROXIMAL PESSIMSITIC FDPO ALGORITHMS

Let $\mathbf{v}_D^{\pi} := \mathcal{E}_{\text{proximal}}(D, \pi)$. From the definition of the proximal family, we have the fixed-point property

$$
\mathbf{v}_D^{\pi} = A^{\pi}(\mathbf{r}_D + \gamma P_D \mathbf{v}_D^{\pi}) - \alpha \left( \frac{\mathrm{TV}_{\mathcal{S}}(\pi, \hat{\pi}_D)}{(1 - \gamma)^2} \right)
$$

and the standard geometric series rearrangement yields

$$
\mathbf{v}_D^{\pi} = (I - \gamma A^{\pi} P_D)^{-1} \left( A^{\pi} \mathbf{r}_D - \alpha \left( \frac{\mathrm{TV}_{\mathcal{S}}(\pi, \hat{\pi}_D)}{(1 - \gamma)^2} \right) \right)
$$

From here, we see:

$$
\begin{aligned}
\mathbf{v}_D^{\pi} &= (I - \gamma A^{\pi} P_D)^{-1} \left( A^{\pi} \mathbf{r}_D - \alpha \left( \frac{\mathrm{TV}_{\mathcal{S}}(\pi, \hat{\pi}_D)}{(1 - \gamma)^2} \right) \right) \\
&= (I - \gamma A^{\pi} P_D)^{-1} A^{\pi} \mathbf{r}_D - (I - \gamma A^{\pi} P_D)^{-1} \alpha \left( \frac{\mathrm{TV}_{\mathcal{S}}(\pi, \hat{\pi}_D)}{(1 - \gamma)^2} \right) \\
&= \mathcal{E}_{\text{naïve}}(D, \pi) - (I - \gamma A^{\pi} P_D)^{-1} \alpha \left( \frac{\mathrm{TV}_{\mathcal{S}}(\pi, \hat{\pi}_D)}{(1 - \gamma)^2} \right)
\end{aligned}
$$

Next, we define a new family of FDPE algorithms,

$$
\underline{\mathcal{E}}_{\text{proximal-full}}(D, \pi) := \mathcal{E}_{\text{naïve}}(D, \pi) - \alpha \left( \boldsymbol{\mu}_{D,\delta}^{\pi'} + (I - \gamma A^{\pi} P_D)^{-1} \left( \frac{\mathrm{TV}_{\mathcal{S}}(\pi, \hat{\pi}_D)}{(1 - \gamma)^2} \right) \right).
$$

We use Lemma 3, which holds with probability at least $1 - \delta$, to bound the overestimation and underestimation. First, the underestimation:

$$
\begin{aligned}
\mathbf{v}_{\mathcal{M}}^{\pi} - \underline{\mathcal{E}}_{\text{proximal-full}}(D, \pi) &= \mathbf{v}_{\mathcal{M}}^{\pi} - \left( \mathcal{E}_{\text{naïve}}(D, \pi) - \alpha \left( \boldsymbol{\mu}_{D,\delta}^{\pi'} + (I - \gamma A^{\pi} P_D)^{-1} \left( \frac{\mathrm{TV}_{\mathcal{S}}(\pi, \hat{\pi}_D)}{(1 - \gamma)^2} \right) \right) \right) \\
&= (\mathbf{v}_{\mathcal{M}}^{\pi} - \mathcal{E}_{\text{naïve}}(D, \pi)) + \alpha \left( \boldsymbol{\mu}_{D,\delta}^{\pi'} + (I - \gamma A^{\pi} P_D)^{-1} \left( \frac{\mathrm{TV}_{\mathcal{S}}(\pi, \hat{\pi}_D)}{(1 - \gamma)^2} \right) \right) \\
&\leq \boldsymbol{\mu}_{D,\delta}^{\pi} + \alpha \left( \boldsymbol{\mu}_{D,\delta}^{\pi'} + (I - \gamma A^{\pi} P_D)^{-1} \left( \frac{\mathrm{TV}_{\mathcal{S}}(\pi, \hat{\pi}_D)}{(1 - \gamma)^2} \right) \right)
\end{aligned}
$$

Next, we analagously bound the overestimation:

$$
\begin{aligned}
\underline{\mathcal{E}}_{\text{proximal-full}}(D, \pi) - \mathbf{v}_{\mathcal{M}}^{\pi} &= \left( \mathcal{E}_{\text{naïve}}(D, \pi) - \alpha \left( \boldsymbol{\mu}_{D,\delta}^{\pi'} + (I - \gamma A^{\pi} P_D)^{-1} \left( \frac{\mathrm{TV}_{\mathcal{S}}(\pi, \hat{\pi}_D)}{(1 - \gamma)^2} \right) \right) \right) - \mathbf{v}_{\mathcal{M}}^{\pi} \\
&= (\mathbf{v}_{\mathcal{M}}^{\pi} - \mathcal{E}_{\text{naïve}}(D, \pi)) - \alpha \left( \boldsymbol{\mu}_{D,\delta}^{\pi'} + (I - \gamma A^{\pi} P_D)^{-1} \left( \frac{\mathrm{TV}_{\mathcal{S}}(\pi, \hat{\pi}_D)}{(1 - \gamma)^2} \right) \right) \\
&\leq \boldsymbol{\mu}_{D,\delta}^{\pi} - \alpha \left( \boldsymbol{\mu}_{D,\delta}^{\pi'} + (I - \gamma A^{\pi} P_D)^{-1} \left( \frac{\mathrm{TV}_{\mathcal{S}}(\pi, \hat{\pi}_D)}{(1 - \gamma)^2} \right) \right)
\end{aligned}
$$

We can now invoke Lemma 1 to bound the suboptimality of any value-based FDPO algorithm which uses $\underline{\mathcal{E}}_{\text{proximal-full}}$, which we denote with $\underline{\mathcal{O}}_{\text{proximal-full}}^{\text{VB}}$. Crucially, note that since $\alpha \boldsymbol{\mu}_{D,\delta}^{\pi'}$ is not dependent on $\pi$, it can be removed from the infimum and supremum terms, and cancels. Substituting and rearranging, we see that with probability at least $1 - \delta$,

$$
\mathrm{SUBOPT}(\underline{\mathcal{O}}_{\text{proximal-full}}^{\text{VB}}(D)) \leq \inf_{\pi} \left( \mathbb{E}_{\rho}[\mathbf{v}_{\mathcal{M}}^{\pi_{\mathcal{M}}^*} - \mathbf{v}_{\mathcal{M}}^{\pi}] + \mathbb{E}_{\rho} \left[ \boldsymbol{\mu}_{D,\delta}^{\pi} + \alpha (I - \gamma A^{\pi} P_D)^{-1} \left( \frac{\mathrm{TV}_{\mathcal{S}}(\pi, \hat{\pi}_D)}{(1 - \gamma)^2} \right) \right] \right)
$$

$$+ \sup_{\pi} \left( \mathbb{E}_{\rho} \left[ \boldsymbol{\mu}_{D,\delta}^{\pi} - \alpha (I - \gamma A^{\pi} P_D)^{-1} \left( \frac{\mathrm{TV}_{\mathcal{S}}(\pi, \hat{\pi}_D)}{(1-\gamma)^2} \right) \right] \right)$$

Finally, we see that FDPO algorithms which use $\underline{\mathcal{E}}_{\text{proximal-full}}$ as their subroutine will return the same policy as FDPO algorithms which use $\underline{\mathcal{E}}_{\text{proximal}}$. First, we once again use the property that $\boldsymbol{\mu}_{D,\delta}^{\pi'}$ is not dependent on $\pi$. Second, we note that since the total visitation of every policy sums to $\frac{1}{1-\gamma}$, we know $\mathbb{E}_{\rho} \left[ (I - \gamma A^{\pi} P_D)^{-1} \left( \frac{1}{\gamma} \right) \right] = \frac{1}{(1-\gamma)^2}$ for all $\pi$, and thus it is also not dependent on $\pi$.

$$\arg\max_{\pi} \mathbb{E}_{\rho}[\mathcal{E}_{\text{proximal-full}}(D, \pi)] = \arg\max_{\pi} \mathbb{E}_{\rho} \left[ \mathcal{E}_{\text{naïve}}(D, \pi) - \alpha \left( \boldsymbol{\mu}_{D,\delta}^{\pi'} + (I - \gamma A^{\pi} P_D)^{-1} \left( \frac{\mathrm{TV}_{\mathcal{S}}(\pi, \hat{\pi}_D)}{(1-\gamma)^2} \right) \right) \right]$$

$$= \arg\max_{\pi} \mathbb{E}_{\rho} \left[ \mathcal{E}_{\text{naïve}}(D, \pi) - (I - \gamma A^{\pi} P_D)^{-1} \alpha \left( \frac{\mathrm{TV}_{\mathcal{S}}(\pi, \hat{\pi}_D)}{(1-\gamma)^2} \right) \right]$$

$$= \arg\max_{\pi} \mathbb{E}_{\rho}[\mathcal{E}_{\text{proximal}}(D, \pi)]$$

Thus, the suboptimality of $\arg\max_{\pi} \mathbb{E}_{\rho}[\mathcal{E}_{\text{proximal}}(D, \pi)]$ must be equivalent to that of $\arg\max_{\pi} \mathbb{E}_{\rho}[\mathcal{E}_{\text{proximal-full}}(D, \pi)]$, leading to the desired result.

## C.9 State-wise Proximal Pessimistic Algorithms

In the main body of the work, we focus on proximal pessimistic algorithms which are based on a state-conditional density model, since this approach is much more common in the literature. However, it is also possible to derive proximal pessimistic algorithms which use state-action densities, which have essentially the same properties. In this section we briefly provide the main results.

In this section, we use $d_{\pi} := (1 - \gamma)(I - \gamma A^{\pi} P_D)^{-1}$ to indicate the state visitation distribution of any policy $\pi$.

**Definition 7.** *A* state-action-density proximal pessimistic algorithm *with pessimism hyperparameter $\alpha \in [0, 1]$ is any algorithm in the family defined by the fixed-point function*

$$f_{\text{sad-proximal}}(\mathbf{v}^{\pi}) = A^{\pi}(\mathbf{r}_D + \gamma P_D \mathbf{v}^{\pi}) - \alpha \left( \frac{|d_{\pi} - d_{\hat{\pi}_D}|}{(1-\gamma)^2} \right)$$

**Theorem 4** (State-action-density proximal pessimistic FDPO suboptimality bound). *Consider any state-action-density proximal pessimistic value-based fixed-dataset policy optimization algorithm $\mathcal{Q}_{\text{sad-proximal}}^{VB}$. Let $\boldsymbol{\mu}$ be any state-action-wise decomposable value uncertainty function, and $\alpha \in [0, 1]$ be a pessimism hyperparameter. For any dataset $D$, the suboptimality of $\mathcal{Q}_{\text{sad-proximal}}^{VB}$ is bounded with probability at least $1 - \delta$ by*

$$\mathrm{SUBOPT}(\mathcal{O}_{\text{sad-proximal}}(D)) \leq 2\mathbb{E}_{\rho}[\boldsymbol{\mu}_{D,\delta}^{\hat{\pi}_D}] + \inf_{\pi} \left( \mathbb{E}_{\rho}[\mathbf{v}_{\mathcal{M}}^{\pi_{\mathcal{M}}^*} - \mathbf{v}_{\mathcal{M}}^{\pi}] + (1 + \alpha) \cdot \mathbb{E}_{\rho} \left[ (I - \gamma A^{\pi} P_D)^{-1} \left( \frac{|d_{\pi} - d_{\hat{\pi}_D}|_+}{(1-\gamma)^2} \right) \right] \right)$$

$$+ \sup_{\pi} \left( (1 - \alpha) \cdot \mathbb{E}_{\rho} \left[ (I - \gamma A^{\pi} P_D)^{-1} \left( \frac{|d_{\pi} - d_{\hat{\pi}_D}|_+}{(1-\gamma)^2} \right) \right] \right)$$

*Proof.* First, note that for any policies $\pi, \pi'$, and any state-action-wise decomposable value uncertainty $\boldsymbol{\mu}$ with state-action-wise Bellman uncertainty $\mathbf{u}_{D,\delta} \leq \frac{1}{1-\gamma} \ddot{\mathbf{1}}$, we have

$$\boldsymbol{\mu}_{D,\delta}^{\pi} - \boldsymbol{\mu}_{D,\delta}^{\pi'} = (I - \gamma A^{\pi} P_D)^{-1} A^{\pi} \mathbf{u}_{D,\delta} - (I - \gamma A^{\pi'} P_D)^{-1} A^{\pi'} \mathbf{u}_{D,\delta}$$

$$= \frac{1}{1 - \gamma} (d_{\pi} - d_{\pi'}) A^{\pi} \mathbf{u}_{D,\delta}$$

$$\leq \frac{1}{1 - \gamma} |d_{\pi} - d_{\pi'}|_+ A^{\pi} \mathbf{u}_{D,\delta}$$

$$\leq \frac{|d_{\pi} - d_{\pi'}|_+}{(1 - \gamma)^2}$$

Invoking Lemma 3 we see that $|\mathbf{v}_D^{\pi} - \mathbf{v}_{\mathcal{M}}^{\pi}| \leq \boldsymbol{\mu}_{D,\delta}^{\pi} \leq \boldsymbol{\mu}_{D,\delta}^{\hat{\pi}_D} + \frac{|d_{\pi} - d_{\hat{\pi}_D}|_+}{(1-\gamma)^2}$. The remainder of the proof follows analogously to Appendix C.8.

$\square$

# D  ALGORITHMS

In the main body of the text, we primarily focus on families of algorithms, rather than specific members. In this section, we provide pseudocode from example algorithms in each family. The majority of these algorithms are simple empirical extensions of well-studied algorithms, so we do not study their properties (e.g. convergence) in detail. The sets of algorithms described here are not intended to be comprehensive, but rather, a few straightforward examples to illustrate key concepts and inspire further research. To simplify presentation, we avoid hyperparameters, e.g. learning rate, wherever possible.

## D.1  NAÏVE

The naïve algorithms presented here are simply standard dynamic programming approaches applied to the empirical MDP construced from the dataset, using $\mathbf{r}_D, P_D$ in place of $\mathbf{r}, P$. See Puterman (2014) for analysis of convergence, complexity, optimality, etc. of the general dynamic programming approaches, and note that the empirical MDP is simply a particular example of an MDP, so all results apply directly.

---

**Algorithm 1:** Tabular Fixed-Dataset Policy Evaluation

**Input:** Dataset $D$, policy $\pi$, discount $\gamma$.
Construct $\mathbf{r}_D, P_D$ as described in Section 2;
$\mathbf{v} \leftarrow (I - \gamma A^\pi P_D)^{-1} A^\pi \mathbf{r}_D$;
**return v**;

---

**Algorithm 2:** Tabular Fixed-Dataset Policy Iteration

**Input:** Dataset $D$, discount $\gamma$.
Initialize $\pi, \mathbf{v}$  Construct $\mathbf{r}_D, P_D$ as described in Section 2;
**while** $\pi$ *not converged* **do**
    $\pi \leftarrow \arg\max_{\dot\pi \in \Pi} A^{\dot\pi}(\mathbf{r}_D + \gamma P_D \mathbf{v})$;   *// argmax policy is state-wise max action*
    $\mathbf{v} \leftarrow (I - \gamma A^\pi P_D)^{-1} A^\pi \mathbf{r}_D$;
**end**
**return** $\pi$;

---

**Algorithm 3:** Tabular Fixed-Dataset Value Iteration

**Input:** Dataset $D$, discount $\gamma$.
Initialize $\pi, \mathbf{v}$  Construct $\mathbf{r}_D, P_D$ as described in Section 2;
**while** $\mathbf{v}$ *not converged* **do**
    $\pi \leftarrow \arg\max_{\dot\pi \in \Pi} A^{\dot\pi} \mathbf{v}$;   *// argmax policy is state-wise max action*
    $\mathbf{v} \leftarrow \mathbf{r}_D + \gamma P_D A^\pi \mathbf{v}$;
**end**
**return** $\pi$;

---

**Algorithm 4:** Neural Fixed-Dataset Value Iteration

**Input:** Dataset $D$, discount $\gamma$.
Initialize $\theta, \theta'$  **while** $\theta$ *not converged* **do**
    **for** *each s in D* **do**
        $\pi(s) \leftarrow \arg\max_{a \in \mathcal{A}} \mathbf{q}_\theta(s, a)$ ;
    **end**
    **while** $\theta'$ *not converged* **do**
        Sample $\langle s, a, r, s' \rangle$ from $D$;
        $L \leftarrow (r + \gamma \mathbf{q}_\theta(s', \pi(s')) - \mathbf{q}_{\theta'}(s, a))^2$;
        $\theta' \leftarrow \theta' - \nabla_{\theta'} L$;
    **end**
    $\theta \leftarrow \theta'$
**end**
**return** $\pi$;

---

## D.2 PESSIMISTIC

In this section, we provide pseudocode for concrete implementations of algorithms in the pessimistic families we have discussed. Since the algorithms are not the focus of this work, we do not provide an in-depth analysis of these algorithms. We briefly note that, at a high level, the standard proof technique used to show convergence of policy iteration (Sutton & Barto, 2018) can be applied to all of these approaches. These algorithms utilize a penalized Bellman update, and the penalty is state-wise and independent between states. Thus, performing a local greedy policy improvement in any one state will strictly increase the value of all states. Thus, policy iteration guarantees strict monotonic improvement of the values of all states, and thus eventual convergence to an optimal policy (as measured by the penalized values).

### D.2.1 UNCERTAINTY-AWARE

In this section, we provide pseudocode for a member of the the UA pessimsitic family of algorithms.

---

**Algorithm 5:** Tabular Uncertainty-Aware Pessimistic Fixed-Dataset Policy Iteration

**Input:** Dataset $D$, discount $\gamma$, error rate $\delta$, pessimism parameter $\alpha$.
Initialize $\mathbf{v}, \pi$  Construct $\mathbf{r}_D, P_D, \ddot{\mathbf{n}}_D$ as described in Section 2;
Compute $\mathbf{u}_{D,\delta}^\pi$ as described in Appendix B;
**while** $\pi$ *not converged* **do**
    $\pi \leftarrow \arg\max_{\mathring{\pi} \in \Pi} A^{\mathring{\pi}}(\mathbf{v} - \alpha \mathbf{u}_{D,\delta}^{\mathring{\pi}})$;
    $\mathbf{v} \leftarrow (I - \gamma A^\pi P_D)^{-1}(A^\pi \mathbf{r}_D - \alpha \mathbf{u}_{D,\delta}^\pi)$;
**end**
**return** $\pi$;

---

**Algorithm 6:** Neural Uncertainty-Aware Pessimistic Fixed-Dataset Value Iteration

**Input:** Dataset $D$, discount $\gamma$, pessimism hyperparameter $\alpha$.
Initialize $\theta, \theta', \Psi$;
**while** $\theta$ *not converged* **do**
    // update policy
    **while** $\Psi$ *not converged* **do**
        Sample $\langle s, \cdot, \cdot, \cdot \rangle$ from $D$;
        $R \leftarrow \mathbb{E}_{a \sim \pi_\Psi(\cdot|s)}[\mathbf{q}_\theta(s,a)] - \alpha \mathbf{u}_{D,\delta}^{\pi_\Psi}(s)$;
        $\Psi' \leftarrow \Psi' + \nabla_{\theta'} R$;
    **end**
    // update value function
    **while** $\theta'$ *not converged* **do**
        Sample $\langle s, a, r, s' \rangle$ from $D$;
        $L \leftarrow \left(r + \gamma\left(\mathbb{E}_{a' \sim \pi_\Psi(\cdot|s')}[\mathbf{q}_\theta(s',a')] - \alpha \mathbf{u}_{D,\delta}^{\pi_\Psi}(s')\right) - \mathbf{q}_{\theta'}(s,a)\right)^2$;
        $\theta' \leftarrow \theta' - \nabla_{\theta'} L$;
    **end**
    $\theta \leftarrow \theta'$
**end**
**return** $\pi_\Psi$;

---

As discussed in the main text, count-based Bellman uncertainty functions, such as those derived from concentration inequalities in Appendix B.1, do not apply to non-tabular environments. The correct way to compute epistemic uncertainty with neural networks is still an open question. Therefore, our pseudocode for a neural implementation of the UA pessimistic approach is in some sense incomplete: a full implementation would need to specify a technique for computing $\mathbf{u}_{D,\delta}^\pi$. We hope that future work will identify such a technique.

### D.2.2 PROXIMAL

In this section, we provide pseudocode for a member of the proximal pessimsitic family of algorithms.

---

**Algorithm 7:** Tabular Proximal Pessimistic Fixed-Dataset Policy Iteration

---

**Input:** Dataset $D$, discount $\gamma$, pessimism parameter $\alpha$.
Initialize $\mathbf{v}, \pi$ Construct $\mathbf{r}_D, P_D, \ddot{\mathbf{n}}_D$ as described in Section 2;
**while** $\pi$ *not converged* **do**

$\quad \pi \leftarrow \arg\max_{\dot{\pi} \in \Pi} A^{\dot{\pi}}(\mathbf{v} - \frac{\alpha}{2(1-\gamma)^2}|\dot{\pi} - \hat{\pi}_D|)$;

$\quad \mathbf{v} \leftarrow (I - \gamma A^\pi P_D)^{-1}(A^\pi \mathbf{r}_D - \frac{\alpha}{2(1-\gamma)^2}|\dot{\pi} - \hat{\pi}_D|)$;

**end**
**return** $\pi$;

---

**Algorithm 8:** Neural Proximal Pessimistic Fixed-Dataset Value Iteration, Discrete Action Space

---

**Input:** Dataset $D$, discount $\gamma$, pessimism hyperparameter $\alpha$.
Initialize $\theta, \theta', \Psi$;
**while** $\theta$ *not converged* **do**

$\quad$ // update policy
$\quad$ **while** $\Psi$ *not converged* **do**

$\quad\quad$ Sample $\langle s, \cdot, \cdot, \cdot \rangle$ from $D$;

$\quad\quad R \leftarrow \mathbb{E}_{a \sim \pi_\Psi(\cdot|s)}[\mathbf{q}_\theta(s, a) - \frac{\alpha}{2(1-\gamma)^2}|\pi_\Psi(s, a) - \hat{\pi}_D(s, a)|]$;

$\quad\quad \Psi' \leftarrow \Psi' + \nabla_{\theta'} R$;

$\quad$ **end**
$\quad$ // update value function
$\quad$ **while** $\theta'$ *not converged* **do**

$\quad\quad$ Sample $\langle s, a, r, s' \rangle$ from $D$;

$\quad\quad L \leftarrow \left(r + \gamma \mathbb{E}_{a' \sim \pi_\Psi(\cdot|s')}\left[\mathbf{q}_\theta(s', a') - \frac{\alpha}{2(1-\gamma)^2}|\pi_\Psi(s', a') - \hat{\pi}_D(s', a')|\right] - \mathbf{q}_{\theta'}(s, a)\right)^2$;

$\quad\quad \theta' \leftarrow \theta' - \nabla_{\theta'} L$;

$\quad$ **end**
$\quad \theta \leftarrow \theta'$
**end**
**return** $\pi$;

---

One important nuance of proximal algorithms is that the optimal policy may not be deterministic, since the penalty term is minimized when the policy matches the empirical policy, which may itself be stochastic. It is not enough to select $\pi_{t+1}(s) = \arg\max_{a \in \mathcal{A}} \mathbf{v}(s, a)$; we must instead select $\pi_{t+1}(\cdot|s) = \sup_{\pi \in \Pi} \mathbf{v}^\pi(s) = \sup_{\pi \in \Pi} \sum_{a \in \mathcal{A}} \pi(a|s)\mathbf{v}(s, a) - \frac{\alpha}{2(1-\gamma)^2}|\pi(a|s) - \hat{\pi}_D(a|s)|$, which is a more difficult optimization problem. Fortunately, it has a closed-form solution.

**Proposition 1.** *Consider any state-action values* $\mathbf{q}$ *and empirical policy* $\hat{\pi}_D$. *Let*

$$z := \max_{a \in \mathcal{A}} \mathbf{q}(\langle s, a \rangle) - \frac{\alpha}{(1-\gamma)^2}$$

. *The policy* $\pi_{localopt}$ *given by*

$$\pi_{localopt}(a \mid s) = \begin{cases} \hat{\pi}_D(a \mid s) + (1 - \sum_{a' \text{ s.t. } \mathbf{q}(\langle s, a' \rangle) > z} \hat{\pi}_D(a' \mid s)) & \text{if } a = \arg\max_{a' \in \mathcal{A}} \mathbf{q}(\langle s, a \rangle), \\ \hat{\pi}_D(a \mid s) & \text{if } \mathbf{q}(\langle s, a \rangle) > z, \\ 0 & \text{otherwise.} \end{cases}$$

*has the property*

$$\sum_{a \in \mathcal{A}} \pi_{localopt}(a|s)\mathbf{q}(\langle s, a \rangle) - \frac{\alpha}{2(1-\gamma)^2}|\pi_{localopt}(a|s) - \hat{\pi}_D(a|s)| \geq \sum_{a \in \mathcal{A}} \pi(a|s)\mathbf{q}(\langle s, a \rangle) - \frac{\alpha}{2(1-\gamma)^2}|\pi(a|s) - \hat{\pi}_D(a|s)|$$

*for all* $s \in \mathcal{S}, \pi \in \Pi$.

*Proof.* We provide a brief outline of the proof. Consider any policy $\pi \neq \pi_{localopt}$ in some state $s$. First, assume that exactly two cells of $\pi(s) - \pi_{localopt}(s)$ are non-zero, meaning that one term is $x$

and the other is $-x$ (since both distributions sum to 1). If $|x| > 0$, the change in penalized return is non-positive, so $\pi$ is worse that $\pi_{\text{localopt}}$. To see this, we simply consider all possible mappings between $x, -x$ and actions. Actions fall into three categories, corresponding to the three cases of the construction: argmax-actions, empirical-actions, and zero-actions, respectively. It's easy to check each case, moving probability mass from any category of action to any other, and see that penalized return is always non-positive. Finally, if more than two cells of $\pi(s) - \pi_{\text{localopt}}(s)$ are non-zero, we can always rewrite it as a sum of positive/negative pairs, and thus the overall change in penalized return is a sum of non-positive terms, and is itself non-positive. $\square$

This closed-form can be used anytime the action space is discrete, in both the tabular and neural proximal algorithms. (In the neural algorithm, it replaces the need to optimize $\Psi$ for a parameterized policy $\pi_\Psi$.)

# E    RELATED WORK

**Bandits.**    Bandits are equivalent to single-state MDPs, and the fixed-dataset setting has been studied in the bandit literature as the *logged bandit feedback* setting. Swaminathan & Joachims (Swaminathan & Joachims, 2015) describe the Counterfactual Risk Minimization principle, and propose algorithms for maximizing the exploitation-only return. The UA pessimistic approach discussed in this work can be viewed as an extension of these ideas to the many-state MDP setting.

**Approximate dynamic programming.**    Some work in the *approximate dynamic programming (ADP)* literature (Munos, 2007) bears resemblance to ours. The setting is very similar; both FDPO and ADP are a modified version of dynamic programming in which errors are introduced, and research on algorithms in these settings studies the emergence and propagation of those errors. The key difference between the two settings lies in the origin of the errors. In ADP, the source of these errors is unspecified. In contrast, in this work, we focus specifically on statistical errors: where they emerge and what their implications are.

**Off-policy evaluation and optimization.**    The problem of off-policy reinforcement learning is typically posed as the setting where the data is being collected by a fixed stationary policy, but an infinite stream of data is being collected, so statistical issues introduced by finiteness of data can be safely ignored. This is clearly closely related to FDPO, but the focus of our work lies precisely on the statistical issues. However, there are close connections; for example, Jiang and Huang Jiang & Huang (2020) show that in the off-policy function approximation setting, algorithms may obey a version of the pessimism principle in order to select a good approximation.

**Exploration.**    Since acting pessimistically is the symmetric opposite of acting optimistically, many papers which study exploration leverage math which is nearly identical (though of course used in an entirely different way). For example, the confidence intervals, modified Bellman equation, and dynamic programming approach of Strehl & Littman (2008) closely mirror those used in the UA pessimistic algorithms.

**Imitation learning.**    Imitation learning (IL) (Hussein et al., 2017) algorithms learn a policy which mimics expert demonstrations. The setting in which these algorithms can be applied is closely related to the FDPO setting, in that IL algorithms map from a dataset of trajectories to a single stationary policy, which is evaluated by its suboptimality. (However, note that IL algorithms can be applied to a somewhat more general class of problems; for example, when no rewards are available.) In the FDPO setting, IL algorithms are limited by the fact that they can never improve on the quality of the empirical policy. In contrast, pessimistic FDPO algorithms will imitate the dataset when other actions are uncertain, but are also sometimes able to deviate and improve.

**Safe reinforcement learning.**    The sub-field of safe reinforcement learning studies reinforcement learning algorithms with constraints or guarantees that prevent bad behavior from occurring, or reduce it below a certain level (Thomas et al., 2019). To this end, many algorithms utilize variants of pessimism. Our contribution distinguishes itself from this line of work via its focus on the

application of pessimism for worst-case guarantees on the more standard objective of expected sub-optimality, rather than for guarantees on safety. However, there is a close relationship between our work and the algorithms and theory used for Safe RL. One popular framework is that of robust MDPs Givan et al. (1997); Nilim & El Ghaoui (2005); Weissman et al. (2003); Iyengar (2005), an object which generalizes MDPs to account for specification uncertainty in the transition functions. This framework is closely related to the pessimistic approaches described in this work. In fact, the UA pessimistic approach is precisely equivalent to constructing a robust MDP from data using concentration inequalities, and then solving it for the policy with the optimal performance under an adversarial choice of parameters; this algorithm has been used as a baseline in the literature but not studied in detail Ghavamzadeh et al. (2016); Laroche et al. (2019). Additionally, the sub-problem of Safe RL known as *conservative policy improvement* focuses on making small changes to a baseline policy which guarantee that the new policy will have higher value Thomas et al. (2015b;a); Ghavamzadeh et al. (2016); Laroche et al. (2019); Simão et al. (2019); Nadjahi et al. (2019), which bears strong resemblance to the proximal pessimistic algorithms discussed in this work.

**Apprenticeship learning.** In the apprenticeship-learning setting (Walsh et al., 2011; Cohen & Hutter, 2020), an agent has the choice to take an action of its own selection, or to imitate a (potentially non-optimal) mentor. The process of determining whether or not to imitate a mentor is similar to determining whether or not to imitate the behavior policy used to collect the data. As a result, the underlying principle of pessimism is relevant in both situations. However, the setting of apprenticeship learning is also different from that of FDPO in several ways (availability of a mentor, ability to collect data), and so prior works do not provide a clear analysis of the importance of pessimism in FDPO.

**Proximal algorithms.** Several approaches to online deep reinforcement learning, including TRPO and PPO, have been described as "proximal" (Schulman et al., 2015; 2017). These algorithms are derived from CPI (Kakade & Langford, 2002), which also introduces a notion of conservatism. In that body of work, this refers to the notion that, after each policy update, the resulting policy is close to the previous iterate. In contrast, the proximal algorithm described in this work is proximal with respect to a fixed policy (typically, the empirical policy defined by the dataset). The contrast between the importance of pessimism in these two cases can be seen by comparing the settings. CPI-type algorithms use pessimism to ensure that the policy remains good throughout the RL procedure, guaranteeing a monotonically-increasing performance as data is collected and the optimal policy is approached. Whereas, FDPO proximal approaches have no guarantees on the intermediate iterates, but rather, use pessimism to ensure that the final policy selected is as close to optimal as possible.

**Deep learning FDPO approaches.** Recently, deep learning FDPO has received significant attention (Agarwal et al., 2019; Fujimoto et al., 2019; Kumar et al., 2019; Laroche et al., 2019; Jaques et al., 2019; Wu et al., 2019; Kidambi et al., 2020; Yu et al., 2020; Wang et al., 2020; Kumar et al., 2020; Liu et al., 2020). At a high level, these works are each primarily focused around proposing and analyzing some specific method. In contrast, the objective of this work is theoretical: we focus on providing a clean mathematical framework through which to understand this setting. We now provide specific details for how our contribution relates to each of these works.

The algorithms introduced in Fujimoto et al. (2019); Laroche et al. (2019); Kumar et al. (2019); Jaques et al. (2019); Liu et al. (2020); Kumar et al. (2020) can all be viewed as variants of the proximal pessimistic approach described in this paper. The implementations vary in a number of ways, but at its core, the primary difference between these algorithms lies in the choice of regularizer: KL, MMD, scalar penalty, or hard constraint. This connection has also been noted in Wu et al. (2019), which provides a unifying algorihtmic framework for these approaches, BRAC, and also performs various empirical ablations. Interestingly, all of these regularizers can be expressed as upper-bounds to $\text{TV}_{\mathcal{S}}(\pi, \hat{\pi}_D)$, and as such, our proof of the suboptimality of proximal pessimistic algorithms can be used to justify all of these approaches. One aspect of our contribution is therefore providing the theoretical framework justifying BRAC. Furthermore, conceptually, these works differ from our contribution due to their focus on error propagation; in contrast, our results show that poor suboptimality of naïve algorithms is an issue even when there is no function approximation error at all.

Wang et al. (2020) provides an alternative algorithm for pessimistic FDPO, utilizing a policy-gradient based algorithm instead of the value-iteration-style algorithms in other related work. Similarly to Kumar et al. (2019), this approach utilizes a policy constraint to prevent boostrapping from actions which are poorly represented in the training data. However, this difference is purely algorithmic: since the fixed-point found by the optimization is the same, this algorithm can also be justified by our proximal pessimistic suboptimality bounds.

Model-based methods (Yu et al., 2020; Kidambi et al., 2020) have recently been proposed which implement pessimism via planning in a pessimistic model. This procedure can be viewed as learning a model which defines an implicit value function (derived from applying the planner to the model), and from there defines an implicit policy (which is greedy with respect to the implicit value function). The implicit value functions learned by the algorithms in Yu et al. (2020); Kidambi et al. (2020) obey the same fixed-point identity as the solutions to the UA pessimistic approach discussed in our work. Thus, both of these works are implementations of UA pessimistic approaches, and our UA pessimistic suboptimality bound can be viewed as justification for this family of algorithms. However, both of these works rely on ensembles to compute the uncertainty penalty, which is not a theoretically well-motivated technique.

The work of Agarwal et al. (2019) provides empirical evidence supporting the claims in this paper. The authors demonstrate that on realistic tasks like Atari, naïve algorithms are able to get good performance given an extremely large and diverse dataset, but they collapse on smaller or less exploratory datasets. It also highlights the difficulties that emerge when function approximation is introduced. Certain naïve algorithms can be seen to still sometimes collapse when datasets are large and diverse, because of other instabilities associated with function approximation.

## F    Experimental Details

### F.1    Tabular

The first set of experiments utilize a simple tabular gridworld. The state space is 8x8, and the action space is {Up, Down, Left, Right}. Rewards are Bernoulli-distributed, with the mean reward for each state-action sampled from Beta(3,1); transitions are stochastic, moving in a random direction with probability 0.2; the discount is .99. This environment was selected to be simple and generic. We compare the performance of four approaches: imitation, naïve, uncertainty-aware pessimistic, and proximal pessimistic. The imitation algorithm simply returns the policy which takes actions in proportion to their observed frequencies in the dataset. For the UA pessimistic algorithm, we use the technique described in Appendix B.1 to implement Bellman uncertainty functions. For both pessimistic algorithms, we absorb all constants into the hyperparameter $\alpha$, which we selected to be $\alpha = 1$ for both algorithms by a simple manual search. For state-actions with no observations, we select $\mathbf{r}_D$ uniformly at random in $[0, 1]$, $P_D$ transitions uniformly at random, and $\hat{\pi}_D$ acts uniformly at random. We report the average of 1000 trials. The shaded region represents a 95% confidence interval.

### F.2    Deep Learning

The second setting we evaluate on consists of four environments from the MinAtar suite (Young & Tian, 2019). In order to derive a near-optimal policy on each environment, we run DQN to convergence and save the resulting policy. We report the average of 3 trials. The shaded area represents the range between the maximum and minimum values.

We implement deep learning versions of the above algorithms in the style of Neural Fitted Q-Iteration (Riedmiller, 2005). In this setting, we implement only proximal pessimistic algorithms. To compute the penalty term, we must approximate $\hat{\pi}_D$; this can be done by training a policy network on the dataset to predict actions via maximum likelihood. Just as in the tabular setting, we absorb all constant coefficients into our pessimism hyperparameter, here setting $\alpha = .25$.

All experiments used identical hyperparameters. Hyperparameter tuning was done on just two experimental setups: Breakout using $\epsilon = 0$, and Breakout using $\epsilon = 1$. Tuning was very minimal, and done via a small manual search.

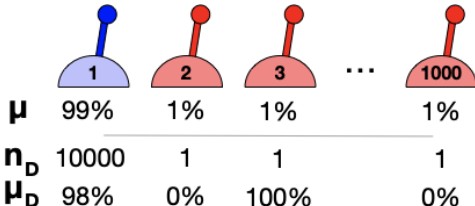

Figure 3: Bandit-like MDP, with accompanying dataset. $\mu$ gives the true mean of each action. $n_\text{D}$ gives the counts of the pulls used to construct dataset $D$, and $\mu_\text{D}$ gives our empirical estimate of the mean reward. On this problem, any algorithm that selects the action with the highest empirical mean reward will almost always pick a suboptimal action. In contrast, a pessimistic algorithm, which selects the action with the highest lower bound, will almost always pick the correct action.

## G   ADDITIONAL CONTENT

### G.1   AN ILLUSTRATIVE EXAMPLE

Consider the following problem setting. We are given an MDP with a single state and some number of actions, each of which return rewards sampled from a Bernoulli distribution on $\{0, 1\}$. (An MDP with a single state is isomorphic to a multi-armed bandit.) Furthermore, for each action, we are given a dataset containing the outcomes of some number of pulls. We are now given the opportunity to take an action, with the goal of maximizing our expected reward. What strategy should we use?

One obvious strategy is to estimate the expected reward for each action by computing its mean reward in the dataset, and then select the arm with the highest empirical reward. However, in certain problem instances, this "naïve strategy" fails. We can illustrate this with a simple example (visualized in Figure 3). Consider an MDP with a single state and 1000 actions. Let the reward distribution of the first action have mean of 0.99, while the reward distributions of all other actions have means of 0.01. Construct a dataset for this bandit by pulling the first arm 10000 times, and the other arms 1 time each.

Although this problem seems easy, the naïve algorithm achieves close to the worst possible performance. It's clear that in this problem, the best policy selects the first action. The empirical estimate of the mean of the first action will be less than 1 with probability $1 - 2 \times 10^{-44}$, and there will be at least one other action which has empirical mean of 1 with probability $1 - 4 \times 10^{-5}$. Thus, the first action is almost never selected.

This issue can be resolved by identifying a fundamental failing of the naïve approach: it ignores epistemic uncertainty around the expected return of each action. In order to guarantee good performance on all problem instances, we need to avoid playing actions that we are uncertain about. One way to do this is to construct a high-probability lower bound on the value of each action (for example using concentration inequalities), and select the action with the highest lower bound. If we consider the upper and lower bounds as defining the set of possible "worlds" that we could be in, acting according to the lower bound of every arm means acting as though we are in the worst possible world. In other words: being pessimistic.

The above example may seem somewhat contrived, due to the enormous size of the action space and skewed data collection procedure. However, we argue that it serves as a good analogy for the more common setting of an MDP with many states and a small number of actions at each state. Roughly speaking, selecting an action in a one-state MDP is analogous to selecting a deterministic policy in a multi-state MDP, and the number of policies is exponentially large in the size of the state space. Additionally, it's very plausible in practical situations that data is collected according to only a small set of similar policies, e.g. expert demonstrations, leading to skewed data coverage.

### G.2  EXTREME PESSIMSIM: VALUE LOWER BOUND ALGORITHMS

We begin with a simple corollary to Lemma 1. Consider any value-based FDPO algorithm whose value function is guaranteed to always return values which underestimate the expected return with high probability.

**Corollary 1** (Value-lower-bound-based FDPO suboptimality bound). *Consider any value-based fixed-dataset policy optimization algorithm $\underline{\mathcal{O}}^{VB}$, with internal fixed-dataset policy evaluation subroutine $\underline{\mathcal{E}}$, which has the lower-bound guarantee that $\underline{\mathcal{E}}(D, \pi) \leq \mathbf{v}_{\mathcal{M}}^{\pi}$ with probability at least $1 - \delta$. For any policy $\pi$, dataset $D$, denote $\mathbf{v}_{D}^{\pi} := \underline{\mathcal{E}}(D, \pi)$. With probability at least $1 - \delta$, the suboptimality of $\underline{\mathcal{O}}^{VB}$ is bounded by*

$$\text{SUBOPT}(\underline{\mathcal{O}}^{VB}(D)) \leq \inf_{\pi} \left( \mathbb{E}_{\rho}[\mathbf{v}_{\mathcal{M}}^{\pi_{\mathcal{M}}^{*}} - \mathbf{v}_{\mathcal{M}}^{\pi}] + \mathbb{E}_{\rho}[\mathbf{v}_{\mathcal{M}}^{\pi} - \mathbf{v}_{D}^{\pi}] \right)$$

*Proof.* This result follows directly from Lemma 1, using the fact that $\mathbf{v}_{D}^{\pi} - \mathbf{v}_{\mathcal{M}}^{\pi} \leq \mathbf{0}$. $\qquad\square$

This bound is identical to the term labeled (A) from Lemma 1. The term labeled (B), which contained a supremum over policies, has vanished, leaving only the term containing an infimum. Where the bound of Lemma 1 demands that some condition hold for all policies, we now only require that there exists any single suitable policy. It is clear that this condition is much easier to satisfy, and thus, this suboptimality bound will typically be much smaller.

A value lower-bound algorithm is in some sense the most extreme example of a pessimistic approach. Any algorithm which penalizes its predictions to decrease overestimation can be described as pessimistic. In such cases, (B) will be decreased, rather than removed entirely. Still, any amount of pessimism reduces dependence on a global condition, decreasing overall suboptimality.

Furthermore, blind pessimism is not helpful. For example, one could trivially construct a value lower-bound algorithm from a naïve algorithm by simply subtracting a large constant from the naïve value estimate of every policy. But this would achieve nothing, as the infimum term would immediately increase by this same amount. To yield a productive change in policy, pessimism must instead vary across states in an intelligent way.

### G.3  PRACTICAL CONSIDERATIONS

Through the process of running experiments in the deep learning setting, the authors noted that several aspects of the experimental setup, which have not been addressed in previous work, had surprisingly large impacts on the results. In this section, we informally report some of our findings, which future researchers may find useful. Additionally, we hope these effects will be studied more rigorously in future work.

The first consideration is that performance is highly nonmonotonic with respect to training time. In almost all experiments, it was observed that with every target-network update, the performance of the policy would oscillate wildly. Even after performing many Bellman updates, few experiments showed anything resembling convergence. It is therefore important that the total number of steps be selected beforehand, to avoid unintentional cherry-picking of results. Additionally, one common trend was for algorithms to have high performance early on, and then eventually crash. For this reason, it is important that algorithms be run for a long duration, in order to be certain that the conclusions drawn are valid.

The second consideration is the degree to which the inner-loop optimization process succeeds. If throughout training, whenever we update the target network, its error is low, convergence near to the fixed point is guaranteed (Antos et al., 2007). However, computational restrictions force us to make tradeoffs about the degree to which this condition is satisfied. Experimentally, we found this property to be very important: when the error was not properly minimized, performance was negatively impacted, sometimes even leading to divergence. There are three notable algorithmic decisions which we found were required to ensure that the error was adequately minimized.

**The size** of the network. It is important to ensure that the network is large enough to reasonably fit the values at all points throughout training. Most prior works (Kumar et al., 2019; Fujimoto et al., 2019; Kumar et al., 2020) utilize the same network architecture as the original DQN (Mnih et al.,

2015), which is fairly small by modern standards. We found that this size of network was adequate to fit MinAtar environments, but that decreasing the size of the network further led to significant performance degradation. Preliminary experiments indicated that since the full Atari environment is much more complex than MinAtar, a larger network may be required.

**The amount of training steps** in the inner loop of Neural Fitted Q-Iteration. If the number of steps is too small, error will not be adequately minimized. In our experiments, approximately 250,000 gradient steps per target update were required to consistently minimize error enough to avoid divergence. We note that many prior works (Kumar et al., 2019; Fujimoto et al., 2019; Kumar et al., 2020) do not adjust this hyperparameter; typical results in the literature use fewer than 10,000 gradient steps per update.

**The per-update initialization** of the network. When changing the target network, we are in essence beginning a new supervised learning problem. Thus, to ensure that we could reliably find a good solution, we found that we needed to fully reinitialize the neural network whenever we updated the target network. The dynamics of training neural networks via gradient descent are still not fully understood, but it is well known that the initialization is of great importance (Hu et al., 2020). It has been observed that certain initializations can seriously impact training: for example, if the final parameters of a classifier trained on randomly-relabeled CIFAR classifier are used as the initialization for the regular CIFAR classification task, the trained network will have worse test-set performance.[9]

---

[9]Chelsea Finn, Suraj Nair, Henrik Marklund; personal communication.

