# OpenReview forum: "The Importance of Pessimism in Fixed-Dataset Policy Optimization"
_ICLR.cc/2021/Conference — ICLR 2021 Poster_

### Official Review · AnonReviewer2 · 2020-10-26
**What are the implications of the decomposition?**

**Rating:** 6
**Confidence:** 4

**Review:**

The message of this paper is that naive policy evaluations common in current (deep) RL algorithms, can lead to a dangerous overestimation of the value function. This overestimation of the value function can then lead to policy improvements with poor theoretical guarantees. To combat overestimation, the authors propose to penalize state-action pairs that are rarely visited. As an easier to implement alternative, and closer to existing algorithms in the literature, the authors also study another penalty term that penalizes deviation from the data generating policy. The authors show on a numerical example that the more principled penalty term that depends on visitation counts is better performing, and that the proximal penalty term only yields minor improvements over imitation learning (i.e. returning the data generating policy).

The main contribution of the paper is to decompose the sub-optimality upper bound into terms that either overestimate or underestimate the total reward that can be collected in the true MDP. The authors argue that the overestimation is especially problematic for (the many) RL algorithms that are subject to such overestimation, as there is a high chance of existence of a policy that performs poorly on the true MDP but has high reward on the empirical MDP (the MDP with empirical estimates of the reward and transitions), resulting in a large sub-optimality.

As far as I am aware, this decomposition is new. But I wonder if beyond formalizing the sub-optimality of naive algorithms, it has other theoretical or practical applications. The notion of pessimism is typical in the analysis of theoretically grounded algorithms (e.g. CPI in
Approximately Optimal Approximate Reinforcement Learning, Kakade et al. 2002), where deviation from ‘known’ state-action pairs is typically maximally penalized with the worst possible value (i.e. a sub-optimality of 1 / (1-\gamma)). So I wonder if the decomposition in overestimation/underestimation terms in Lemma 1 allows for new theoretical insights and algorithmic developments or if it is more of a rewriting, and similar results can be obtained by more carefully choosing the empirical MDP D, such that the overestimation term disappears with high probability even in the worst case and only the underestimation term remains. As is, I understand the reasons for exhibiting both underestimation/overestimation terms in order to analyze ‘naive’ algorithms in the sense of Sec. 4, but is there an advantage for this decomposition and for the algorithm in Sec. 5.1 compared to choosing the optimal policy without an penalty term but in a more carefully constructed MDP D’ that doesn’t allow for overestimation? Similarly, is there any benefit in not choosing \alpha = 1 in Sec. 5.1? Is there an optimal choice for \alpha for Sec. 5.2?

As for the practical implications, the results in Sec. 6 are quite depressing since algorithms with a proximal penalty are easier to implement than with the uncertainty penalty. What was the \alpha in the experiments? I wonder if the results can be improved for the proximal algorithm if a better choice of \alpha is used depending on the optimality of the data generating policy or on the size of the dataset.

Overall, the paper is an interesting read and its message is well presented and supported. However, I am wondering if the theoretical contributions can serve another purpose than warning about the poor theoretical guarantees of ‘naive’ algorithms, and hope the authors can correct me if I underappreciated the importance of these derivations.

---

> ### Author Response · Authors · 2020-11-17
> **Response, pt 1**
>
> Thank you for your comments. Your main concerns seem to be around the relevance of the results; to clarify the relevance, we have re-organized and rewritten some sections of the paper in order to emphasize the impactful insights. Here, we also will attempt to briefly explain.
>
> As you identified, the core contribution of our work is around the importance of pessimism in this setting: mathematically characterizing the specific reason that being pessimistic yields improvements. So, why is understanding the reason useful? Here is our perspective:
>
> In the past few months, there has been an enormous amount of interest in developing algorithms for FDPO. (See “Deep learning FDPO approaches” in Appendix E for a list of recent related work.) Our contribution serves to inform and motivate the design of pessimistic algorithms for this setting. Whereas other works have mostly proposed solutions, we explain *why* some solutions are better than others.
>
> One key takeaway from our work, which you correctly identified, is that proximal pessimistic algorithms are in some sense a special case of uncertainty-aware algorithms. A huge proportion of recently-proposed algorithms have been proximal algorithms, but all have the same fundamental limitations. One clear reason our research is valuable is that it conclusively demonstrates that to move beyond, we will need to research uncertainty-aware algorithms. (We have now made this connection more explicit in a new section, 5.3 of the revised draft.)
>
> Here are three further examples of how our work can be used to gain insight into recent algorithms, and how it can serve to inform and guide future research.
> - Wang et al. (2020) propose a new algorithm which uses policy-gradient to find a solution instead of Q-learning; however, since we see from our analysis that its performance is dependent on its fixed-point (which is not impacted by the search procedure), it will have the same suboptimality as any other proximal algorithm.
> - Kumar et al. (2020) propose a new algorithm which they claim improves performance because it guarantees a Q-function lower bound; however, our work shows that it is simply a variant of a proximal algorithm, and the lower-bound property is not relevant to suboptimality.
> - Liu et al. (2020) propose a new algorithm which is proximal to the state-distribution of the empirical policy, rather than the state-action distribution; our analysis reveals that this is still simply a slight variant of proximal algorithms, and its suboptimality will have similar properties.

---

> > ### Author Response · Authors · 2020-11-17
> > **Response, pt 2**
> >
> > To respond to some of the other specific points you raised:
> > - *Connection to CPI.* Yes, this is a very insightful connection. The setting of CPI is quite different from the setting we consider, in that CPI involves constantly collecting more data, so it only needs to stay proximal to the most recent policy. (In contrast to our work, which stays proximal to the behavior policy on a fixed dataset). CPI’s choice of 1/(1-γ) as a penalty is precisely analogous to the choice made by the proximal algorithms we discuss, and indeed, it implements a form of pessimism. Our work indicates that there is likely an “uncertainty-aware CPI”, which is able to be more sample-efficient while retaining the key property of safe monotonic improvement, by using non-trivial uncertainty estimates (instead of the trivial one of 1/(1-γ). This is an interesting direction for future work.
> > - *On a “pessimistic empirical MDP” formulation.* The intuition you describe is a good one, but we believe our fixed-point formulation is better than a pessimistic-MDP formulation for two concrete technical reasons. Firstly, a pessimistic empirical MDP is much less general, and doesn't allow you to derive some of the more interesting approaches. This is clear by simply noting that any pessimistic empirical MDP is still an MDP, and thus has a deterministic optimal policy, whereas the proximal & 'hedging' approaches (Appendix B.2) both have stochastic optimal policies, and as such cannot be represented. (The more general “bounded-parameter MDP” aka “robust MDP” framework, e.g. Givan et al. 2002, is general enough, but it introduces significant analytical complexity relative to our approach.) The second reason is that, although our results were primarily tabular, one aim of this work is to develop a mathematical framework that can easily be extended to the deep learning setting. Many popular deep learning algorithms are based on Q-learning, and the penalty formulation aligns much better with the structure of these algorithms than a pessimistic-MDP formulation.
> > - *Similarly, is there any benefit in not choosing \alpha = 1 in Sec. 5.1? Is there an optimal choice for \alpha for Sec. 5.2?* Yes. In general the optimal value of alpha that minimizes the bound is neither 0 nor 1.  Since the supremum term will typically be much larger than the infimum term on real-world environments, it is typically the case that we will want an α much closer to 1. However, it’s not clear how to compute the true optimal α, which is data- and environment-dependent.
> > - *What was the \alpha in the experiments? I wonder if the results can be improved for the proximal algorithm if a better choice of \alpha is used depending on the optimality of the data generating policy or on the size of the dataset.*  In our experiments, hyperparameters were intentionally selected arbitrarily and tuned minimally, to ensure that the qualitative results were not cherry-picked. The tabular experiments used a hyperparameter of α=.25. Our experiments are designed to test empirically whether algorithms have the properties predicted by our theoretical results. For both proximal and UA algorithms, changing alpha from 0 to 1 gives an interpolation from the naive curve to the imitation curve, and at intermediate values (which includes our selected values), curves look similar to the results we presented. It’s likely that for any specific problem, performance can be improved somewhat by hyperparameter tuning α, but there is no value of α that will make the proximal curve look like the UA curve.
> > Also, note that information on “the optimality of the data generating policy” is not in general available in the FDPO setting, so we cannot use it to tune. Tuning on “the size of the dataset” is possible, but does not lead to a principled algorithm -- see our response to Question 1 from Reviewer 1. Finally, note as well that any tuning which requires testing out policies in the real environment is not permitted by the FDPO setting.

---

> > > ### Comment · AnonReviewer2 · 2020-11-20
> > >
> > > Thank you for your thorough response.  Regarding '*Our work indicates that there is likely an “uncertainty-aware CPI”, which is able to be more sample-efficient while retaining the key property of safe monotonic improvement*' you might be interested in Safe Policy Iteration, Pirotta et al. 2013. I think it goes into a direction of a more refined and easy to implement pessimism than 1/(1-$\gamma$), although it is not in the FDPO setting.
> > >
> > > Of course, it would have been better if the authors arrived at a concrete solution to close the gap between the naive algorithms with loose theoretical guarantees and the theoretical algorithms that are hard to implement. But I agree with R1 that the derivations of a unified framework might drive future researcher in this direction so I am raising my score accordingly.

---

### Official Review · AnonReviewer3 · 2020-10-27
**needs better organization and some missing related work**

**Rating:** 6
**Confidence:** 4

**Review:**

Summary:

The paper proposes a theoretical framework for analyzing the error of reinforcement learning algorithms in a fixed dataset policy optimization (FDPO) setting.  In such settings, data has been collected by a single policy that may not be optimal and the learner puts together a model or value function that will have explicit or implicit uncertainty in areas where the data is not dense enough.  The authors provide bounds connecting the uncertainty to the loss.  They then show that explicitly pessimistic algorithms that fill in the uncertainty with the worst case can minimize the worst case error.  Similarly, proximal algorithms that attempt to adhere to the collection policy (as often the case in model-free batch RL) have improved error compared to a naive approach but not as good as an explicitly pessimistic approach.


Review:

The paper provides a general description of the pessimism performance bounds.  The theorems appear to be correct and the reasoning sound.  I also like the connection to the proximal approach, which is how most model-free batch RL algorithms approach the problem (by sampling close to the collection policy).

However, the paper does need some improvement.  Specifically, a connection should be made to more existing literature on pessimism in safe, batch, or apprenticeship RL.  In addition, the paper spends a lot of time on definitions and notation that are not explicitly used while the most interesting empirical results are relegated to the appendix, which seems backwards.

On the connections to the literature, the idea of using pessimism in situations where you are learning from a dataset collected by a non-optimal teacher has been investigated in previous works in apprenticeship RL:
http://proceedings.mlr.press/v125/cohen20a/cohen20a.pdf
or
https://papers.nips.cc/paper/4240-blending-autonomous-exploration-and-apprenticeship-learning.pdf

Specifically, the first (Bayesian) paper explicitly reasons about the worst of all possible worlds mentioned in the current submission and seems to have a lot of overlap in the theory.  Can the authors distinguish their results from Cohen et al.?  The second paper is an example where model-learning agents keep track of the uncertainty in their learned transition and reward functions and use pessimism to fill in uncertainty.  So the idea here is not quite new and better connections to this literature need to be made.

The other issue with the paper is its organization and writing. The theoretical results, while general, are not particularly complicated and don’t seem to warrant the amount of notation and definitions on pages 1-3.  Specifically, the bandit example isn’t really mentioned in the paper but the figure takes up a lot of valuable space.  Over a full page is used to define basic MDP and dataset terms that are widely known and commonly used.  The footnotes are whole paragraphs that seem to be just asides.  Finally, the grid word results are presented in a figure without any real associated text except for some generalities about what algorithms worked well,  Meanwhile, the most interesting and novel contributions of the paper, including the concrete algorithms for applying pessimistic learning, and the empirical analysis on Atari games, are stashed in the (very long) appendix.  I strongly suggest the authors reorganize the paper to highlight these strengths instead of notation and footnotes that are tangential to the paper.

---

> ### Author Response · Authors · 2020-11-17
> **Response**
>
> Thank you for the feedback. Your major concerns seem to be the organization of the paper and connection to related work.
>
> At the suggestion of you and the other reviewers, we have updated the draft to have improved organization by shrinking the notation & background section (moving most details to the appendix), and moving much of the discussion from the appendix to the main body of the paper. We’ve posted the latest revision -- hopefully this clarifies the presentation to your satisfaction.
>
> Thank you as well for suggesting these references; we were not previously familiar with the apprenticeship learning literature, and have added it to the related works in Appendix E. In response to your request, we’d be happy to specifically distinguish our work from Cohen at al. Though the intuitions are similar, the specific technical differences from Cohen et al. are quite significant. Firstly, as mentioned previously, the setting is different: the apprenticeship learning setting, as defined in that work, involves the problem of continually observing a mentor and deciding whether to imitate (defer) or take an alternative action. This process of trading off the collection of new data with the use of old knowledge has much more in common with active learning or reinforcement learning than it does with FDPO, which is pure exploitation. Next, their solution implicitly relies on a good prior; while all of our analysis is prior-free. Their key results are around convergence to the quality of the mentor, whereas ours are around the reduction of suboptimality. Also, their approach is considerably more abstract, and less amenable to implementation; their discussion centers around an “idealized agent”, writing, “...this agent is only tractable when the model class is very simple, but it can inspire tractable approximations”. In contrast, our algorithms are concrete and implementable, and we in fact implement them, in both the tabular and deep learning settings. Thus, the work of Cohen et al. is related conceptually, but complimentary when it comes to technical content.
>
> Regarding the related work at a high level: we wish to emphasize that our core contribution is not “pessimism”, in general, but rather a technical analysis of the importance of pessimism in the *specific* context of FDPO. (We've updated the language in the introduction of the paper to make this more clear.) FDPO is also known as batch RL or offline RL, however, it is somewhat different from the apprenticeship setting (where a teacher can be queried for additional data) and the safety setting (several formulations, but in general, the objective is some notion of safety rather than minimizing suboptimality). For readers who are interested in any of these other, related settings, we provide a comprehensive overview of all of these connections in Appendix E. Are there any particular connections which you feel are central enough that they need to be included in the main text?

---

### Official Review · AnonReviewer1 · 2020-10-28

**Rating:** 7
**Confidence:** 4

**Review:**

**Summary:**

This paper attempts to unify prior work on fixed-dataset (aka "batch" or "offline") reinforcement learning. Specifically, it emphasizes the importance of pessimism to account for faulty over-estimation from finite datasets. The paper shows that naive algorithms (with no pessimism) can recover the optimal policy with enough data, but do so more efficiently. The pessimistic algorithms are divided into "uncertainty-aware" and "proximal" algorithms where the uncertainty-aware algorithms are shown to be more principled, but most prior work falls into the computationally easier proximal family of algorithms that is closer to imitation learning. These insights are proven both theoretically and with some small experiments.

--------------------------------------------------------------------

**Strengths:**

1. A nice decomposition of suboptimality. The main workhorse of the paper is the decomposition provided in Lemma 1 which is novel and can provide some good intuition about the necessity of pessimism (although the intuition is only given in appendix G.3, which should definitely find it's way into the main text). The Lemma cleanly and formally demonstrates why we may expect over-estimation to be more damaging than under-estimation.
2. A clear framework to examine prior work. The paper does well to capture the majority of recent work into a few broad families of algorithms: naive, proximal pessimistic, and uncertainty-aware pessimistic. The bound derived from the main Lemma for each algorithm family provide evidence to prefer uncertainty-aware algorithms. This is supported by the tabular experiments.
3. The formal statements of Lemmas and Theorems seem to be correct and experimental methodology seems sound.

--------------------------------------------------------------------

**Weaknesses:**

1. I am wary of the comparison of upper bounds done in the paper. Just because one algorithm has a lower upper bound does not prove superior performance. I agree that since all the proofs are derived from Lemma 1 and are very similar, the differences are indeed suggestive. However, the bound in Theorem 3 seems to be more loose than the others. For example, when $\alpha = 0$ it does not recover the bound for the naive algorithm as would be expected. A more measured tone and careful description of these comparisons is needed. Claims like "uncertainty-aware algorithms are strictly better than proximal algorithms" in the conclusion are not substantiated.
2. Lack of discussion of issues with implementation and function approximation. As the authors get into in Appendix G.6 and Appendix F.2 and briefly in the paper it is not clear how to implement the uncertainty-aware family of algorithms in a scalable way. I am not saying that this paper needs to resolve this issue (it is clearly hard), but this drawback needs to be made more clear in the main text of the paper, so as to not mislead the reader.
3. Notation is heavy and sometimes nonstandard. I understand that the nature of this paper will lead to a lot of notation, but I think the paper could be made more accessible if the authors go back through the paper and remove notation that may only be needed in the proofs and may be unnecessary to present the main results. For example, the several different notions of uncertainty funtions might be useful in the appendix, but do not seem to all be necessary to present the main results. Similarly, the notion of decomposability is introduced and then largely forgotten for the rest of the paper. Some notation is nonstandard. For example: $d$ is used for number of datapoints (usually it would be dimension) and $ \Phi$ is used as the data distribution (usually if would be a feature generating function or feature matrix).
4. Abuse of the appendix. While I understant that the 8 page limit can be difficult, this paper especially abuses the appendix often sending important parts of the discussion and intuition for the results into appendix G. The paper would be stronger with some editing of the notation and organization of the main text to make room for more of the needed discussion and intuition in main body of the paper.

--------------------------------------------------------------------

**Recommendation:**

I gave the paper a score of 7, and recommend acceptance. The paper provides a nice framing of prior work on fixed-dataset RL. While it leaves some things to be desired in terms of carefulness, scalability, and clarity, I think it provides a solid contribution that will be useful to researchers in the field.

If the authors are able to sufficiently improve the clarity of presentation as discussed in the weaknesses section, I could consider raising my score.

--------------------------------------------------------------------

**Questions for the authors:**

1. It is natural to think that a practical proximal pessimistic algorithm would reduce the level of pessimism with the dataset size (so that it approaches the naive algorithm with infinite data). Do approaches like this resolve many of the issues that you bring up with proximal pessimistic algorithms (albeit by introducing another hyperparameter to tune)?

--------------------------------------------------------------------

**Additional feedback:**

Typos:

- The first sentence on page 4 is not grammatically correct.
- In the statements of Lemma 1 and Theorem 1, $ \pi^*_D$ is defined and never used.
- In the statement of Theorem 1 $ u_{D,\delta}^\pi$ is defined but then only $ \mu_{D,\delta}^\pi$ is used without being defined.

---

> ### Author Response · Authors · 2020-11-17
> **Response**
>
> Thank you for the thorough feedback. It seems your major concern, raised in weaknesses 2-4, is the organization of the paper. Specifically, you disagreed with the choice to include certain things (notation), while leaving others in the appendix (discussion). We agree. The additional notation is needed for the proofs, but indeed not necessary to the main paper since the proofs are in the appendix.
>
> At the suggestion of you and the other reviewers, we have updated the draft to have improved organization by shrinking the notation & background section (moving most details to the appendix), and moving much of the discussion from the appendix to the main body of the paper. We’ve posted the latest revision -- hopefully this clarifies the presentation to your satisfaction.
>
> We also wish to respond to the other two points you raised.
>
> Weakness 1:
> The justification for the claim "uncertainty-aware algorithms are strictly better than proximal algorithms" was somewhat buried in the appendix, but it truly does hold rigorously. In the latest draft, we’ve added a new section, 5.3, which makes this point clear. Here’s the argument at a high level: proximal algorithms are equivalent to a special case of uncertainty-aware algorithms, where we use the trivial state-action-wise Bellman uncertainty function which returns 1/(1-γ) everywhere. Thus, not only are uncertainty-aware algorithms strictly more general than proximal algorithms, but proximal algorithms are in a certain sense the “worst” algorithm of this type. As a result, we believe that our strong phrasing in the conclusion is warranted.
>
> You also raise a very good point about the fact that the suboptimality of a proximal algorithm with α=0 does not recover the original bound for the proximal algorithm. This is due to a choice of presentation of the result, not a fundamental weakness of the analysis. There are two ways we could write the result. The tighter way is: inf (A + αB) + sup (A - αB), where A is the term from the naive analysis. In this case, we’d recover the desired result at α=0. But, since B >= A, we instead chose to write it as: inf (1+α)B + sup (1 - α)B, which emphasizes the fact that the term vanishes when α=1. However, we agree that losing the equivalence at α=0 is a real problem, and as such, have switched it to the tighter version. See the updated Theorem 3 in the revised draft for the updated result.
>
> Question 1:
> The approach you propose would make the algorithm consistent, but still run into problems in the finite data regime. Consider what happens if you take a dataset D, choose one transition <s,a,r,s’>, and create a new dataset D’ := D + (1e100 * <s,a,r,s’>), i.e., add 1e100 identical copies of the transition. Consider running your algorithm on D’. Since the size of D’ is greater than 1e100, the pessimism scaling factor (which, per your description, shrinks with dataset size) will be close to 0, meaning we will simply be using the naive algorithm. But of course, adding 1e100 copies of the same transition has not informed us about any other state-actions, so naive suboptimality will be very poor. Thus, the suboptimality of your algorithm on this problem will be poor as well.

---

### Decision · Program_Chairs · 2021-01-07
**Final Decision**

**Decision:**

Accept (Poster)

**Comment:**

The reviewers agree in their positive evaluation of the paper. A weakness of the paper pointed out by several reviewers was its presentation, which has hovewer improved. Thus, I'm glad to recommend acceptance.